



# Classification of North Atlantic and European extratropical cyclones using multiple measures of intensity

Joona Cornér[1], Clément Bouvier[1], Benjamin Doiteau[2,3], Florian Pantillon[2], and Victoria A. Sinclair[1]

[1]Institute for Atmospheric and Earth System Research / Physics, Faculty of Science, University of Helsinki, Finland
[2]Laboratoire d'Aérologie, Université de Toulouse, CNRS, UPS, IRD, Toulouse, France
[3]CNRM, Université de Toulouse, Météo-France, CNRS, Toulouse, France

**Correspondence:** Joona Cornér (joona.corner@helsinki.fi)

**Abstract.** The question of how to quantify the intensity of extratropical cyclones does not have a simple answer. To offer some perspective on this issue, multiple measures of intensity are analyzed in this study for North Atlantic and European extratropical cyclones for the extended winter season between 1979 and 2022 using ERA5 reanalysis data. The most relevant intensity measures are identified by investigating relationships between them and by performing sparse principal component analysis on the set of measures. We show that dynamical intensity measures correlate strongly with each other, while correlations are weaker for impact-relevant measures. Based on the correlations and the sparse principal component analysis, we find that five intensity measures, namely 850 hPa relative vorticity, 850 hPa wind speed, wind footprint, precipitation, and a storm severity index, describe cyclone intensity comprehensively and non-redundantly. Using these five measures as input, we objectively classify the extratropical cyclones with a cluster analysis based on a Gaussian mixture model. The cluster analysis is able to produce four clusters between which cyclones differ in terms of their intensity, life cycle characteristics such as deepening rate and lifetime, and geographical location. A clear majority ($81\%$) of investigated impactful storms belong to the most intense cluster, which demonstrates the ability of the method to identify potentially damaging extratropical cyclones.

## 1 Introduction

Extratropical cyclones (ETCs), also frequently referred to as mid-latitude cyclones or low pressure systems, constitute a substantial part of the atmospheric circulation in the mid-latitudes and transport large amounts of heat, moisture, and momentum polewards (Hartmann, 2015). ETCs are also responsible for most of the day-to-day variability in sensible weather in the mid-latitudes and are the dynamical cause for most of the precipitation (Hawcroft et al., 2012). Furthermore, the most extreme ETCs can be associated with heavy precipitation and strong winds responsible for flooding, landslides, damage to infrastructure and buildings, or diverse economic losses.

No two ETCs are the same and there is great variability in their shape, size, lifetime, and intensity (Nielsen and Dole, 1992). Thus, many attempts of classifications have been made and have often been driven by the desire to better understand the development or the structure of certain types of ETCs. For example, Zillman and Price (1972) and Browning (1990) classified ETCs based on their cloud patterns, whereas Field and Wood (2007) grouped ETCs based on their low-level wind speed and their water vapour path in an attempt to understand ETC precipitation. Attempts have also been made to classify ETCs by their



dynamical forcing. For example, Thorncroft et al. (1993) and Schultz et al. (1998) separated ETCs that develop in different background flows, whereas Petterssen and Smebye (1971), Deveson et al. (2002), and Dacre and Gray (2009) grouped ETCs that are dominated either by low-level thermal advection or upper-level vorticity advection. However, perhaps the most common approach to group or classify ETCs is based on a measure of their intensity. Previous studies have grouped ETCs either by their maximum 850 hPa relative vorticity (VO), often focusing on the strongest storms (e.g., Catto et al., 2010; Sinclair et al., 2020),

or by the decrease in minimum mean sea level pressure (MSLP) over a 24-hour period, again commonly focusing on the most rapidly deepening storms (e.g., Sanders and Gyakum, 1980; Reboita et al., 2021).

Quantifying the intensity of any given ETC in a concise yet accurate manner is however a challenging task. Meteorologists and climate scientists have faced this challenge for many decades and as such many methods and diagnostics exist, but there is no clear "correct" way or "best" diagnostic. The most common metrics used to quantify ETC intensity, MSLP and VO, describe

the synoptic-scale dynamics of the ETCs and are strongly related to the horizontal pressure gradient and large-scale winds. Historically, MSLP has been provided by records from surface stations, while nowadays VO is also commonly available from large gridded datasets such as reanalyses. However, in many cases, neither the minimum MSLP nor maximum VO correlates well with the impacts of a given ETC. This is mainly due to the presence of mesoscale features such as fronts, low-level jets and convergence bands which strongly influence the wind and precipitation fields (Hewson and Neu, 2015). Therefore, impact-

relevant metrics which have no direct theoretical link to the traditional dynamically based metrics of MSLP and VO have been introduced. Such metrics include precipitation rates and accumulations (Hawcroft et al., 2012), sizes of wind footprints (Roberts et al., 2014), and storm severity indices (Leckebusch et al., 2008a).

Though challenging, quantifying the intensity of ETCs is crucial for climate studies, given the large role these weather systems play in the climate system. Firstly, many studies have used reanalysis datasets to quantitatively describe the state of the

current climate in terms of number, location, and intensity of ETCs (e.g., Hoskins and Hodges, 2002; Rudeva and Gulev, 2007; Jeglum et al., 2010; Laurila et al., 2021a). Secondly, concise metrics are needed to identify whether any trends in ETCs intensity have already occurred or may do in the future as the climate changes. For example, the Coupled Model Intercomparison Project (CMIP) has been extensively analysed to determine how the intensity of ETCs may change in the future (e.g., Zappa et al., 2013b; Colle et al., 2013; Seiler and Zwiers, 2016; Chang, 2018). In multiple CMIP models and in four different emission

scenarios, Priestley and Catto (2022) find that in the Northern Hemisphere winter, the most extreme cyclones have higher VO, lower MSLP, stronger wind speeds, and extreme wind speeds covering a larger area in the future compared to present day, whereas changes are minimal for average cyclones. In simulations performed with the Community Earth System Model Large Ensemble (CESM-LE) for North Atlantic ETCs, Dolores-Tesillos et al. (2022) find small changes for VO, a slight increase in the maximum wind speeds, and an increase in the size of wind footprints. Thirdly, concise metrics of ETC intensity enable

the comparison of representation of ETCs in different datasets. For example, ETC climatologies can differ between different reanalyses because the ETCs differ in the underlying model used to produce them, in the data assimilation methods, or in their spatial and temporal resolution (Wang et al., 2016). Furthermore, by comparing ETCs in historical simulations to ETCs in reanalysis datasets, it is possible to determine how accurately CMIP models reproduce the current climate (Catto et al., 2010; Priestley et al., 2020).



Up to now, most studies have only quantified ETC intensity using a single metric, or, if more than one has been considered, they have been used independently of each other. This may result in some information being omitted, as each metric only gives information about one aspect of ETCs. For example, two ETCs with the same minimum MSLP may have considerable different wind gusts associated with them, or be very different in size (Sinclair, 1997). Considering a vast number and diversity of measures would likely give an in-depth description of all ETCs, but this would become impractical to deal with, hard to visualize, and challenging for forecasters and researchers to easily comprehend. Therefore, an optimal balance should be

sought. In this context, automated and objective methods need to be applied to group ETCs using more than one metric or the spatial variation in a variable. One such method is *k*-means clustering (Lloyd, 1982), which has been applied extensively in atmospheric sciences and also in particular to cluster ETCs. This method has been based, for example, on the large-scale synoptic structure at genesis time (Catto, 2018), the track orientation (Wang et al., 2024), or the precipitation patterns associated

with ETCs (Sinclair and Catto, 2023). Other cluster analysis methods have also been applied, including probabilistic methods such as regression mixture models (Gaffney et al., 2007).

The purpose of the study is to classify ETCs using multiple measures of intensity. The first aim is to identify how a number of commonly used ETCs intensity measures relate to each other, and then to identify the optimal metrics which, when considered together, are fully describing the intensity of ETCs. The second aim is to classify the wintertime ETCs in the North Atlantic

and in the European region based on this subset of intensity measures, and to quantify the characteristics of each cluster. The last aim of this study is to show where some of the previously studied high-impact ETCs occur in our phase space of intensity and ETCs classification.

The paper is structured as follows. Section 2 explains how the dataset of ETC tracks and intensity measures was created. Section 3 describes the methods used in the analysis of the data. Section 4 contains the results of the study and Section 6

concludes and discusses them.

## 2   Creation of ETC intensity dataset

### 2.1   ERA5 reanalysis

Reanalyses are considered to be the best estimate of the past state of the atmosphere. They are available globally, with high spatio-temporal resolution and long-term coverage, which makes them essential tools for studying phenomena with large

spatial and temporal scales. ERA5 (Hersbach et al., 2020) is a global reanalysis dataset provided by the European Centre for Medium-range Weather Forecasts (ECMWF). It is produced with model forecasts of CY41R2 of the ECMWF's Integrated Forecast System (IFS) and 4D-Var data assimilation. The original ERA5 dataset consists of 137 model levels, however, data are also provided on 37 pressure levels. It has a horizontal spectral resolution of $T_1639$, which corresponds to a grid spacing of $0.28°$/31 km at the equator on the native regular Gaussian grid of ERA5. ERA5 data are available from 1940 until the present

in near real time, with a temporal resolution of one hour. However, we only utilize data from 1979 onwards since ERA5 data after 1979 is considered more reliable thanks to the availability of satellite observations assimilated.



We use ERA5 reanalysis as it is an open access and easy-to-use dataset which has been used extensively in previous mid-latitude cyclone research (e.g., Karwat et al., 2022; Dacre et al., 2023). ERA5 has been found to perform well in representing many variables (Hersbach et al., 2020, and references therein). For example, Molina et al. (2021) found reasonable represen-

tation of 10 m wind speed in Europe, and Lavers et al. (2022) showed that the smallest errors in precipitation are found in the extratropics during winter. However, like all reanalyses, ERA5 has some biases which are products of the model formulation and physical parameterizations as well as the observing system and data assimilation scheme. For example, Chen et al. (2024) evaluated ERA5's ability to correctly represent the 10 m wind speed and precipitation in the presence of ETCs in North America. They find that overall ERA5 represents 10 m winds well with a small overall bias, but overestimates weak winds and

underestimates strong winds. Precipitation is less well captured than winds, and high values of precipitation are considerably underestimated. Minola et al. (2020) also find that ERA5 underestimates strong wind speeds and strong wind gusts in Sweden under various weather conditions. Despite some biases, we use ERA5 for all variables to have dynamical consistency between all considered variables, spatial uniformity, and a high resolution in the data.

ERA5 data are used to both create ETC tracks (Sect. 2.2) and extract the ETC intensity measures (Sect. 2.3). The data are

used on pressure levels and selected surface fields, and are taken with a time resolution of three hours. The focus in this study is on the North Atlantic storm track during the extended Northern Hemisphere winter season from October to March in the 1979–2022 period.

## 2.2  ETC tracking

ETC tracks are identified with the objective feature tracking software TRACK (Hodges, 1994, 1995, 1999b). TRACK uses

a Lagrangian approach of tracking individual cyclones by identifying extrema in a given field and following them through time. To track ETCs, we use VO like many other studies have previously done (e.g., Catto et al., 2011; Catto, 2018; Sinclair et al., 2020). The benefits of using VO instead of MSLP include less influence from the background environment, and the fact that ETC tracks are often found earlier in their life cycle (Hoskins and Hodges, 2002). Additionally, unlike VO, MSLP is an extrapolated field which is affected by the extrapolation method and the representation of orography in the model. A drawback

of tracking with VO is that the ETC identification is not reliable in areas where the climatological value of surface pressure is close to or less than 850 hPa. However, this is the case only in areas of high orography, such as Greenland and the Alps.

To track ETCs, the VO field at the native horizontal resolution of ERA5 is first truncated to T42 spectral resolution ($310\,\mathrm{km}$ at the equator) to exclude small scale features and ensure that only synoptic-scale ETCs are identified. Wave numbers less than five are also removed to filter out planetary-scale waves. For the remaining wave numbers, local maxima are identified

in the filtered VO field and a nearest neighbour approach is used to connect them into ETC tracks. The smoothest tracks are identified by minimizing a cost function (Hodges, 1994). Finally, TRACK produces output which consists of the horizontal location (longitude and latitude) and magnitude of the T42 VO maxima for each time step in each ETC track.

We further filter the ETC tracks using the following criteria:



1. To exclude stationary and short-lived systems, the tracks need to be at least $1000\,\mathrm{km}$ long and last for at least two days (16 time steps).

2. Weak systems are excluded by using a minimum threshold of $1 \times 10^{-5}\,\mathrm{s}^{-1}$ for the T42 VO.

3. Like in Sinclair and Catto (2023), the maximum T42 VO of the track must occur 24 hours after genesis (the first time step) or later.

4. The maximum T42 VO needs to occur inside the area $80°$ W to $40°$ E in longitude and $30°$ N to $75°$ N in latitude (the magenta box in Fig. 1).

In total, in the 43 extended winters we find 7361 tracks with these criteria.

Statistical diagnostic fields of ETC tracks, such as track density, are calculated by using spherical kernel estimators provided by TRACK (Hodges, 1996, 1999a, 2008). Figure 1 shows the climatology of track density of the 7361 ETC tracks. At a given grid cell, track density is defined as the number of ETC tracks per season occurring within a spherical cap with a radius of five geodesic degrees. In Fig. 1 we see that track density is the largest along the North Atlantic storm track, beginning from the eastern coast of North America and extending northeastward towards Northern Europe. A local maximum in track density can also be seen in the Mediterranean basin. This is despite the fact that the tracking algorithm we use and the filters we applied are more designed to identify ETCs in the main storm tracks than in the Mediterranean, where ETCs tend to be smaller and shorter-lived (Campins et al., 2011). Similar distributions of track density during winter (DJF) were identified previously by Priestley et al. (2020) in historical CMIP6 simulations, and by Hoskins and Hodges (2002) in ECMWF analyses using the same tracking algorithm. Using MSLP from ERA5 as input to a different tracking algorithm, Feser et al. (2021) also found a similar distribution in the North Atlantic. Campins et al. (2011), Aragão and Porcù (2022), and Doiteau et al. (2024), who focused on the Mediterranean region, determined the Gulf of Genoa as the location of maximum track density during DJF, whereas we have a maximum in the Tyrrhenian Sea. This difference can be explained by the sensitivity of the tracking method in this particular basin (see Flaounas et al. (2023) for details). However, the overall number of tracks in the Mediterranean area in these previous studies is in agreement with our distribution.

## 2.3  Intensity measures

We use ERA5 reanalysis data to create a set of intensity measures for the tracked ETCs. We reduce the amount of data to contain one value per intensity measure per track. All the intensity measures can be calculated from any other reanalysis dataset.

The intensity measures can be divided into two categories. The first category consists of dynamical measures which describe physical aspects of ETCs and are available as is in the reanalysis. The second category consists of what we call impact-relevant measures. These are diagnostic variables which are designed to quantify the aspects of ETCs that possibly have societal impacts. We emphasize that the impact-relevant measures do not necessarily directly translate into impacts, but are based on variables which quantify the ETC features that have been found to cause the most damage to infrastructure.





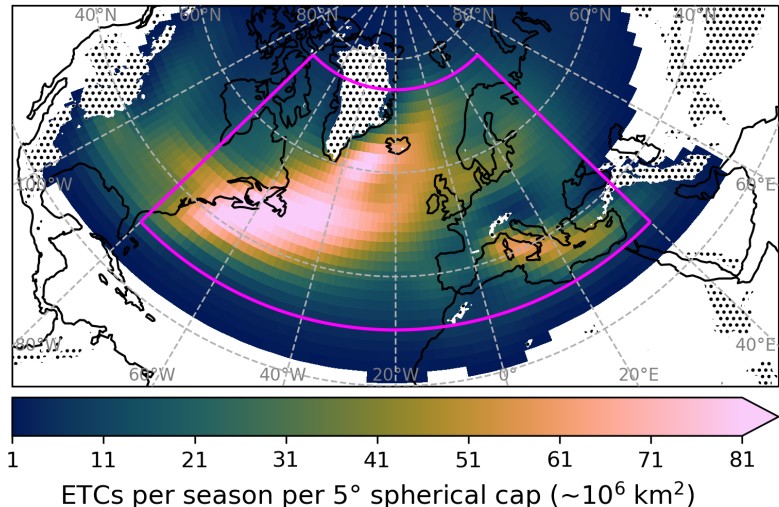

**Figure 1.** Climatology of ETC track density for the 43 ONDJFM seasons in areas where on average at least one track per $5°$ spherical cap ($\sim 10^6\,\mathrm{km}^2$) occurs per season. The magenta box bounded by $80°$ W, $40°$ E, $30°$ N and $75°$ N shows the area inside which the ETCs need to have their maximum T42 VO. The hatching indicates areas where the average monthly mean value of surface pressure between October 1979 and March 2022 is below $850\,\mathrm{hPa}$ and the tracks may therefore be non-physical.

All intensity measures and the details of how they are produced are described in detail in the following subsections (Sect. 2.3.1 and Sect. 2.3.2), and the measures are summarized in Table 1. Most of the variables are on the native ERA5 grid with the horizontal resolution of $0.28°$. The exceptions are 850 hPa wind speed and 10 m wind speed, which are on a $0.25°$ regular longitude–latitude grid. This is solely because we already had this data locally available. However, this difference in resolution is small and very unlikely to have an effect on the results.

### 2.3.1 Dynamical measures

The baseline for the intensity measures is VO, a widely-used measure of ETC intensity. We use VO directly from the output of TRACK. For the sake of simplicity, we refer to the location of the VO maximum as the ETC centre in the following, although the physical centre of the ETC is not necessarily located exactly at the same point.

The next dynamical variable we include is based on MSLP, which, along with VO, is a widely used measure of ETC intensity. For example, Priestley et al. (2020) used VO and MSLP to assess future changes in ETC intensity. To account for the effect of the background environment on the MSLP values(i.e., large-scale temporal variations and climatological dependence of latitude (Anderson et al., 2003) ), we subtract a monthly mean value from the MSLP field to obtain an MSLP anomaly (MSLPa). We use the MSLPa field to find the nearest local minimum around the VO maximum. This is done by using bilinear spline interpolation and a steepest-descent method with TRACK. The local minimum MSLPa is found inside a circle centered around the VO maximum and has a radius of six geodesic degrees (equivalent to about $670\,\mathrm{km}$). Various values were tested for





the radius, and six geodesic degrees was discovered to be the most suitable (not shown). This value is also used by e.g. Li et al. (2014). Most of the MSLPa minima are within $100\,\mathrm{km}$ of the VO maxima, and very few values are actually found at a distance of $670\,\mathrm{km}$ (see Fig. S1). For a small number of maximum VO values, TRACK is unable to find an associated MSLPa value.

Finally, we include winds from multiple levels as dynamical intensity measures. For a comprehensive overview of winds

associated with ETCs, we include the maximum wind speed at 850 hPa (WS850), 925 hPa (WS925) and 10 m (WS10) within six geodesic degrees from the VO centre, a radius also used by Zappa et al. (2013a) and Gramcianinov et al. (2020). We also include the wind gusts at 10 m (FG10) as the maximum within six geodesic degrees, but unlike for the other three wind variables, we do not use an instantaneous value. Wind gust is highly variable in time and therefore we take the maximum value of the previous three hours. As for MSLPa, various values were explored for the radius of the area for all wind speed measures

(not shown).

### 2.3.2 Impact-relevant measures

The first impact-relevant intensity measure is based on precipitation which is an hourly accumulated field in ERA5, i.e., the value gives the rate of precipitation during the previous hour. Because of this, we sum precipitation values over the three previous hours to get a precipitation rate which corresponds to our time interval. Based on this three-hour precipitation rate we

define a precipitation diagnostic (PRECIP) as the average precipitation rate, $P_{\mathrm{ave}}$, within an area, using a definition adapted from Sinclair and Catto (2023):

$$P_{\mathrm{ave}} = \frac{1}{A_{\mathrm{T}}} \sum_{i=1}^{m} P_i A_i \tag{1}$$

where $P_i$ is the precipitation rate in grid point $i$, $A_i$ is the area of grid point $i$, $m$ is the number of grid points within a certain geodesic radius from the ETC centre in which the precipitation rate exceeds a specific threshold value, and $A_{\mathrm{T}}$ is the total

area of the $m$ grid points. Thus, precipitation is considered and averaged only over the area where it exceeds the threshold. Sinclair and Catto (2023) used a radius of $12°$ on a spherical grid and a minimum precipitation rate of $1\,\mathrm{mm}\,(6\,\mathrm{h})^{-1}$. We use the same values adapted to our grid and time resolution, i.e., a geodesic radius of $12°$ and a minimum precipitation rate of $0.5\,\mathrm{mm}\,(3\,\mathrm{h})^{-1}$.

Next, we construct a wind footprint diagnostic to measure the area of the ETC wind field. The wind footprint (WFP) is

defined as:

$$\mathrm{WFP} = \sum_{i=1}^{m} A_i \tag{2}$$

where $A_i$ is the area of grid point $i$ and $m$ is the number of grid points within a certain geodesic radius from the ETC centre in which the maximum 10 m wind gust values within 3 hours exceed a specific threshold. We use a value of $10°$ geodesic for the radius. This radius was found to be the best compromise between capturing winds associated with a given ETC from an

area as large as possible, without contaminating the WFP with winds related to neighbouring ETCs (see Fig. S2). For the wind gust threshold, we use a relatively small value of $15\,\mathrm{m}\,\mathrm{s}^{-1}$ to have non-zero WFP for moderate ETCs as well. The threshold



value for wind gust was chosen by considering values used by various national weather services in Europe to issue the lowest level (yellow) wind warnings over land. In Finland this value is $15\,\mathrm{m\,s^{-1}}$ in summer and $20\,\mathrm{m\,s^{-1}}$ in winter (FMI), whereas the corresponding values in Norway are $17\,\mathrm{m\,s^{-1}}$ and $19\,\mathrm{m\,s^{-1}}$ (METNorway). In Ireland this value is $25\,\mathrm{m\,s^{-1}}$ (MetÉireann)
whereas in Germany the threshold is $14\,\mathrm{m\,s^{-1}}$ (DWD). The small wind gust value selected here is also justified based on the fact that in ERA5, wind gust may be underestimated in some areas (Chen et al., 2024; Minola et al., 2020).

Finally, we include a storm severity index (SSI) to quantify the possible impact from wind associated with ETCs. Different SSI metrics have been previously used to successfully estimate the societal impact of ETCs (Klawa and Ulbrich, 2003; Pinto et al., 2007; Leckebusch et al., 2007). We use here an SSI metric adapted from Leckebusch et al. (2008a). The SSI is calculated
in each grid point within a circular area with a fixed radius (in geodesic degrees) around the ETC centre throughout the ETC life cycle, whereas Leckebusch et al. (2008a) considered SSI only in time steps during which SSI is non-zero in a minimum threshold area. Our definition of SSI is as follows:

$$\mathrm{SSI} = \frac{1}{A_{\mathrm{ref}}} \sum_{i=1}^{m} \max(0, \frac{v_i}{v_{98,i}} - 1)^3 A_i, \tag{3}$$

where $v_i$ is the maximum 10 m wind gust within 3 hours in grid point $i$, $v_{98,i}$ is the climatological 98th percentile value of
10 m wind gust in grid point $i$, $A_i$ is the area of grid point $i$, $m$ the number of grid points within a certain geodesic radius from the ETC centre, and $A_{\mathrm{ref}}$ is the area of the largest point in the grid (at the equator). Scaling SSI values with the relative area of each grid point ensures that with the same wind gust exceedance, SSI values are higher in larger grid cells than smaller grid cells. In other words, the storm is considered more severe because the impacted area is larger. This also means that at higher latitudes where grid points have smaller area and thus there are more of them within the radius, the SSI values are on average
smaller, leading to more consistent values independent of latitude.

The climatology of 10 m wind gust is calculated for the whole period from October 1979 to March 2022, including the summer months as well. We include summer months in the climatology, since the justification for using the 98th percentile as a threshold for the wind gust is based on the finding that damage from winds occurs locally on $2\,\%$ of all days (Palutikof and Skellern, 1991). In some regions (e.g., parts of Scandinavia, in the Mediterranean, or Southeastern Europe) the climatological
98th percentile values of 10 m wind gust are quite small (Fig. S3) and unlikely to cause severe damage. Karremann et al. (2014) avoided this discrepancy between the definition of SSI and actual wind gust values by implementing a fixed minimum threshold ($9\,\mathrm{m\,s^{-1}}$ for 10 m wind speed). For impactful events in the Mediterranean region, Nissen et al. (2010) required a minimum duration of 18 hours and minimum affected area of around $36000\,\mathrm{km^2}$. We do not use a minimum wind gust or area threshold for the SSI, since such an approach is already evaluated by the WFP, which is based on a fixed gust threshold. We
use a geodesic radius of $10°$ as the maximum radius of the area also for the SSI, a value which was discovered to be the most suitable based on a similar analysis as was performed for the WFP (see Fig. S4).

For each ETC track, we restrict the analysis to only one time step per intensity measure. For all other intensity measures other than PRECIP, the value at the time of maximum VO along the track is chosen, since the maximum value of the intensity measures also occurs at the same time or at the adjacent time step on average (see Fig. S5). For PRECIP, the chosen time is 12
hours before the time of maximum VO, for the same reason.





We also include accumulated versions of PRECIP and SSI. We construct an accumulated precipitation measure (PRECIPacc) by summing together the PRECIP values and an accumulated SSI measure (SSIacc) by integrating the SSI values w.r.t. time along each whole track.

| Measure | Abbreviation | Type | Distance | Time step |
|---|---|---|---|---|
| 850 hPa relative vorticity | VO | Maximum T42 | 0 | 0 |
| Mean sea level pressure anomaly | MSLPa | Nearest local minimum | 6 | 0 |
| 850 hPa wind speed | WS850 | Maximum | 6 | 0 |
| 925 hPa wind speed | WS925 | Maximum | 6 | 0 |
| 10 m wind speed | WS10 | Maximum | 6 | 0 |
| 10 m wind gust | FG10 | Maximum | 6 | 0 |
| Precipitation | PRECIP | Avg. where $\geq 0.5\,\mathrm{mm}\,(3\,\mathrm{h})^{-1}$ | 12 | -12 |
| Accumulated precipitation | PRECIPacc | Avg. where $\geq 0.5\,\mathrm{mm}\,(3\,\mathrm{h})^{-1}$ | 12 | Accumulated |
| Wind footprint | WFP | Area where gust $\geq 15\,\mathrm{m\,s}^{-1}$ | 10 | 0 |
| Storm severity index | SSI | Sum over area | 10 | 0 |
| Accumulated storm severity index | SSIacc | Sum over area | 10 | Time-integrated |

**Table 1.** Summary of all 11 intensity measures. The columns are the full name, abbreviated name, the type of value extracted, distance from ETC centre (in geodesic degrees), and time step w.r.t. the time of maximum vorticity (in hours), respectively. The horizontal line separates the measures into dynamical and impact-relevant ones.

## 3 Methods

### 3.1 Correlation analysis

We use two different correlation metrics to quantify co-occurrence relationships between the intensity measures. The first one is the widely-used Pearson's correlation coefficient, $r$, to evaluate linear dependence. The second one is mutual information (MI) which we use to quantify non-linear dependencies. MI is a measure of dependence between two random variables based on their joint and marginal entropies (Cover and Thomas, 2006). In other words, it quantifies how much information can be obtained about one random variable by observing another random variable. MI has values in the interval $[0,\infty)$ but it can be converted into a correlation coefficient, $\rho$, by normalizing it to a range $[0,1]$. The normalization is computed using the Python package ennemi (Laarne et al., 2021, 2022) as follows:

$$\rho = \sqrt{1 - \exp(-2\mathrm{MI})}. \tag{4}$$

Using two different kinds of correlation metrics is beneficial, because in addition to quantifying the strength of the relationship, we can learn about its type as well. Because MI is able to quantify non-linear relationships in addition to linear ones,





similar values of $r$ and $\rho$ indicate a linear relationship between two variables, whereas higher $\rho$ than $r$ values mean that the relationship has a non-linear component.

## 3.2 Principal component analysis

Principal component analysis (PCA) is a statistical method which aims to find an orthogonal linear transformation that maxi-
mizes the variance projected onto each of the newly found axes. This method is heavily used in all fields of science, including meteorology (Statheropoulos et al., 1998; Nagendra and Khare, 2003), and has also been applied in studies on cyclones (Lou et al., 2012; Nakajo et al., 2014; Chen et al., 2019). One of the main features of PCA is its ability to reduce the dimensionality of a given problem by ignoring the axes explaining a negligible amount of the original variance. However, one of its main drawback is the interpretability of the results as each axis is expressed by a linear combination of the original feature space, of-
ten mobilizing the entire space. Sparse PCA (sPCA) alleviates this issue by proposing a PCA with sparse loadings, i.e., setting some of the coefficients in the linear expression of the PCA's axes to zero (Zou et al., 2006; Zou and Xue, 2018). However, sPCA is a lossy compression technique with an important dependency on the dimension of the projective hyperplane.

Consequently, in this study we will guide the sPCA by a classical PCA analysis (respectively scikit-learn's SparsePCA and PCA (Pedregosa et al., 2011)). We apply the PCA to estimate the explained variance against the number of principal
components (PC). Then, we select the number of components which explains more than 90 % of the variance. This number of components is then used as the main parameter of the sPCA and is applied to the same dataset. Therefore, we minimize the risk of losing too much information with PCA by conserving a satisfying interpretability with sPCA.

## 3.3 Cluster analysis

Based on the correlations and the sPCA results, we select a reduced subset of intensity measures to use as input for the cluster
analysis. This subset is conceived to reduce information redundancy whilst maintaining the interpretability of the original set of measures. As this set is created to provide a relevant and compact description of ETCs' intensities, it can also be used as an input for a machine learning analysis.

Machine learning (ML) methods have gained popularity in recent years due to their predictive and classification abilities, especially for ETCs (Watanabe et al., 2020; Sinclair and Catto, 2023; Kurth et al., 2023). Of the many ways to classify
meteorological datasets, unsupervised learning (i.e., clustering) is often prioritized to group elements of the dataset without any *a priori* knowledge. The Gaussian mixture model (GMM) is a popular clustering method which aims to fit several multivariate Gaussian distributions to a dataset. As such, each cluster may be represented as a multidimensional Gaussian probability density function extending throughout the whole feature space, in our case the small subset of intensity measures produced by the sPCA analysis. The main drawback of GMM is that the number of clusters has to be input, and the optimal number of
clusters cannot be known in advance. The elbow method is used to disambiguate this choice and select the number of clusters which 1) maximize Silhouette score and 2) do not fall in the over-fitting learning plateau (i.e., when the Silhouette score is constant). In other words, the optimal number of clusters is the case avoiding both under- and over-fitting.





We use the GMM (scikit-learn's GaussianMixture (Pedregosa et al., 2011)) on the reduced subset of intensity measures selected by our sPCA method (cf. Sect. 3.2). To choose the correct number of clusters to be found, we first use the elbow method to select two values to test before the over-fitting plateau is reached, as shown in Fig. S6. Then, we use the following stability test to refine our decision to one value. Our stability test aims to verify if the clusters' centroids predicted by several instances of the GMM are intercomparable. We first compute the Euclidean distance between the reference centroids and the centroids predicted by 1000 other instances. Then, the arguments of the minimum of the Euclidean distances are taken for each predicted clusters. If the arguments of the minimum are not repeating, it means a permutation of the clusters is able to successfully compare two instances of GMM. In other words, we test the sensitivity of our cluster analysis to the chosen number of clusters. A stability score has been defined as the average number of clusters which are not intercomparable between two instances of GMM. Thus, a stability score of zero is considered optimal. Stability scores are shown in Table S1. As a result, we select $n = 4$ as our optimal number of clusters as this value falls in the elbow criteria and has the lowest stability score.

## 4 Results

### 4.1 Relationships between intensity measures

Distributions of all 11 of the intensity measures are shown in Fig. 2. We see in Fig. 2a–f that the dynamical intensity measures have Gaussian-like distributions. Distributions of VO, MSLPa and WS850 (Fig. 2a–c) are slightly skewed towards more intense values. The positive skewness is to be expected for VO because of the requirement for a minimum threshold value. The VO values range from $1.5 \times 10^{-5}\,\mathrm{s}^{-1}$ to around $17 \times 10^{-5}\,\mathrm{s}^{-1}$. A similar distribution was found by Bengtsson et al. (2006) and Zappa et al. (2013a). For MSLPa there is no fixed threshold value, but the largest minimum values can be expected to be around $0\,\mathrm{hPa}$, i.e., below the climatological average MSLP. Therefore, the distribution is skewed towards negative values. The most negative values are around $-80\,\mathrm{hPa}$ while the most positive values are around $20\,\mathrm{hPa}$. WS925, WS10, and FG10 have more symmetric distributions (Fig. 2d–f) than VO, MSLPa, or WS850. WS850, WS925, and FG10 have similar variances, with the mean value being around $30\,\mathrm{m\,s}^{-1}$ in all three. WS10 has a smaller variance than the other three wind speed measures and a mean value of around $20\,\mathrm{m\,s}^{-1}$. Gramcianinov et al. (2020) found a comparable distribution of WS10 and Bengtsson et al. (2009) of WS925.

Compared to the dynamical intensity measures, the distributions of the impact-relevant measures are much less Gaussian-like (Fig. 2g–k). Out of these, PRECIP has the most Gaussian-like distribution (Fig. 2g). It is however more positively skewed than any of the dynamical measures' distributions. PRECIP values range from $0\,\mathrm{mm}\,(3\,\mathrm{h})^{-1}$ to around $6\,\mathrm{mm}\,(3\,\mathrm{h})^{-1}$. Zappa et al. (2013b) found a similar distribution with a slightly different definition for precipitation. The distribution of PRECIPacc (Fig. 2h) is even more positively skewed than that of PRECIP. The differences between ETCs are emphasized with the added effect of the duration of the track. PRECIPacc values range from about $10\,\mathrm{mm}$ to more than $700\,\mathrm{mm}$. The same effect of the minimum threshold, as is the case for VO, can thus be seen in the precipitation distributions. WFP (Fig. 2i) has a large peak at the smaller end of the distribution and is very flat almost until the largest values. The first bin in the histogram contains ETCs with WFP values below $0.2 \times 10^6\,\mathrm{km}^2$, which is equal to the area of a square with a side length of about $440\,\mathrm{km}$. Only around



ETCs have a WFP of zero (not shown), which means that the first bin has the most ETCs with non-zero WFP. The WFP values decrease rapidly at the large end of the distribution as the radius threshold is reached (the theoretical maximum value of WFP is around $3.9 \times 10^6 \, \text{km}^2$). The shapes of the SSI distributions are even more extreme (Fig. 2j–k, shown on semi-log axes). The cube of the wind exceedance in Equation (3) for SSI emphasizes differences between small and large values. Therefore,

there are many very small values and few large values. For example, there are outlier ETCs in which the values of SSI and SSIacc are more than twice the value of the next largest one. These ETCs are the 1993 Storm of the Century (Huo et al., 1995) and Ex-hurricane Wilma (Pasch et al., 2006), respectively.

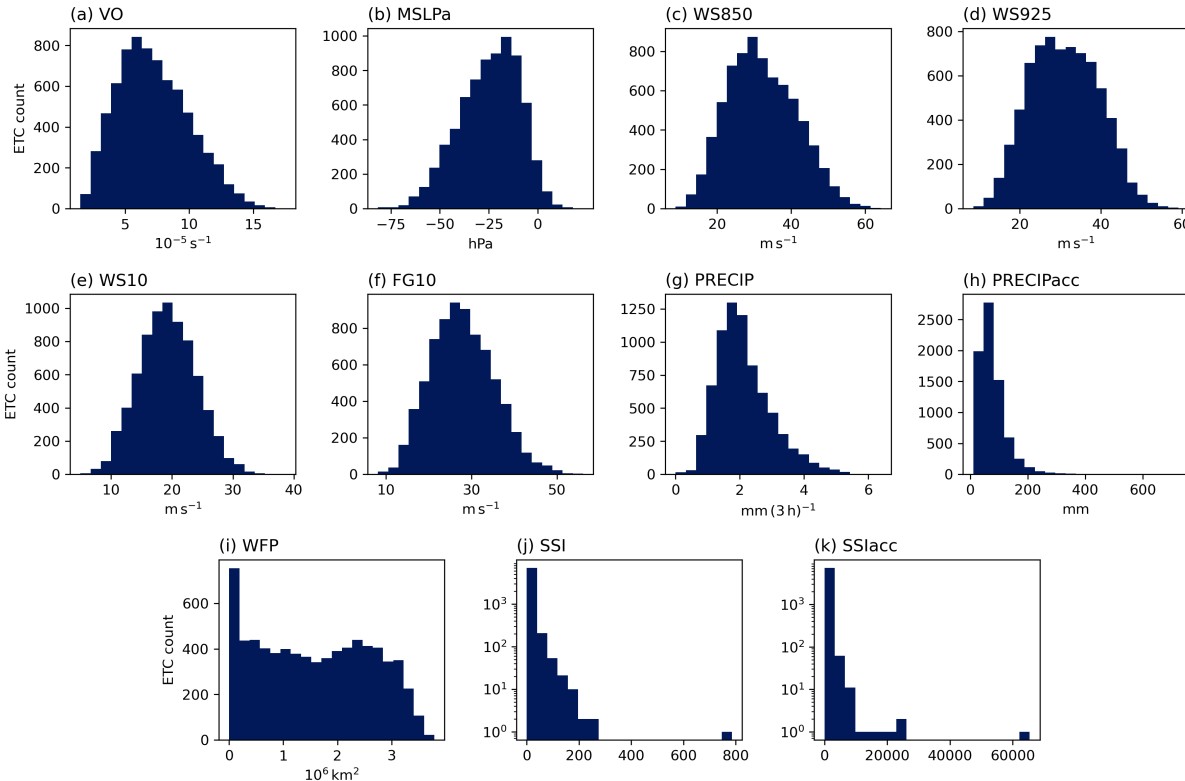

**Figure 2.** Distributions of intensity measures at selected times in the tracks (see text for details).

We investigate the Pearson correlation coefficients ($r$) in Fig. 3a and correlation coefficients from MI ($\rho$) in Fig. 3b for relationships between all 11 intensity measures. The relationships can be roughly divided into two groups based on their

strength: the first group contains the dynamical measures and WFP, and the second one the SSI and precipitation measures. All the Pearson correlations are statistically significant (not shown).

Correlations between all measures in the group consisting of the dynamical measures and WFP are strong. All the relationships in this group have a Pearson's $r$ of at least 0.7 (Fig. 3a). The strongest correlations are between the four wind speed measures. A particularly strong correlation is between WS10 and FG10 with a Pearson's $r$ of 0.97. This is surprising given





that in the IFS, the parameterization for wind gusts includes a term to represent the contribution of convective downdrafts and a term to account for surface roughness in addition to the 10 m wind speed term (Bechtold and Bidlot, 2009). The very strong correlation between WS10 and FG10 found here indicates that the convective downdrafts and surface friction contribute a minimal amount to the wind gusts in ETCs in ERA5. The weakest correlations in the first group of measures are between MSLPa and the wind speed measures, with $r$ values between 0.71 and 0.76. Correlation coefficients from MI (Fig. 3b) are

consistently, yet only slightly, larger than Pearson's $r$ for the measures in the first group. The small difference between the two correlation coefficients suggests that the relationships are linear.

     Relationships between the other measures are weaker, as well as between SSI and precipitation measures. This can be explained by the different shapes of the distributions in Fig. 2. For both SSI and precipitation measures, the largest values of Pearson's $r$, around 0.5, are between the corresponding accumulated and instantaneous versions. For any of these four

measures, the strongest correlation with another measure is between precipitation (both PRECIP and PRECIPacc) and VO with a value of $r = 0.47$, which, for PRECIP, may be related to the selection of the maximum precipitation value 12 hours before the time of maximum VO. This may be explained by, e.g., precipitation-related diabatic heating producing a low-level potential vorticity anomaly which feeds back to the 850 hPa relative vorticity (Davis and Emanuel, 1991). Correlations between SSI and precipitation measures are the weakest in terms of $r$ and among the weakest in terms of $\rho$.

For the SSI and precipitation measures, the $\rho$ values are consistently larger than $r$ values, and this difference is larger than for measures in the first group. The relationships are thus more non-linear. However, the correlation coefficients from MI are still not as high as for the first group, with the strongest correlation between FG10 and SSI having $\rho = 0.71$. Coefficients increase more for SSI measures than precipitation measures, to values of around $0.6 \pm 0.1$. This is not unexpected given the highly non-Gaussian distribution of the SSI measures.

**4.2   PCA and Sparse PCA**

Figure 4 shows the result of the PCA with the 11 intensity measures as input. By investigating the weights of the intensity measures in the first four PCs (Fig. 4b–e), we see that there are multiple measures with a contribution of similar magnitude in each PC. For example, in the first PC (Fig 4b) WFP has the largest weight but VO, MSLPa, and all wind speed measures have similar, non-negligible weight. In Fig. 4a, which shows ETCs projected onto the first three PCs of the PCA space, this

can be seen in the even spread of points all over the axes. Based on the result of the PCA, it is not straightforward to determine which intensity measures have redundancy between them and which do not. Therefore, it is difficult to use only the PCA for dimensionality reduction in the original dataset.

     However, we can use the fact that the first four PCs of the PCA contain $94\,\%$ of explained variance in the dataset to constrain the sPCA. The result of the sPCA constrained to four PCs with the 11 intensity measures as input is shown in Fig. 5. As

opposed to the result of the PCA, now each of the four PCs consists almost completely of either a single intensity measure or a group of similar intensity measures. We see this in their larger weight compared to the other measures, which have a weight close to or exactly zero in the same PC (Fig. 5b–e). The PCs can therefore be labelled as consisting of 1) the four dynamical wind speed measures (WS850, WS925, WS10, and FG10), 2) PRECIP, 3) WFP, and 4) VO and MSLPa. Minor contributions





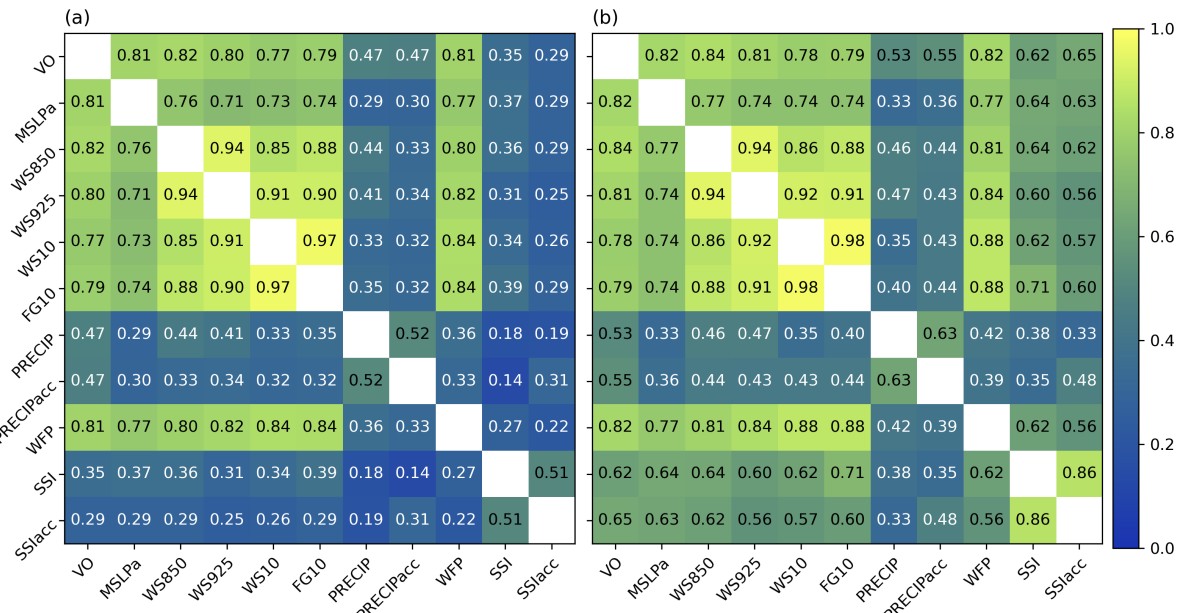

**Figure 3.** (a) Pearson and (b) MI correlation coefficients for the ETC intensity measures. An absolute value of Pearson's $r$ is shown for correlations involving MSLPa to aid comparison with other coefficients.

in terms of weight come from 1) VO, 2) PRECIPacc and VO, 3) WS10 and FG10, and 4) WS850. The only measures which have no weight in any of the PCs are the two SSI measures.

The PCs now have a straightforward physical interpretation, and we can label the axes in the sPCA space according to ETC features quantified by the most important intensity measures in the PCs (e.g., windiness). In Fig. 5a the ETCs are shown projected onto the first three PCs of the sPCA space in which PC1 goes from "calm" to "windy", PC2 goes from "dry" to "rainy", and PC3 goes from "small" to "big". For reference, the most "average" ETC in the sPCA space (the smallest Euclidean distance from origin) has a WS850 value of $30.2\,\mathrm{m\,s^{-1}}$, PRECIP of $2.3\,\mathrm{mm\,(3\,h)^{-1}}$, WFP of $1.7 \times 10^6\,\mathrm{km^2}$, and VO of $6.8 \times 10^{-5}\,\mathrm{s^{-1}}$. Compared to the projection of the PCA space in Fig. 4a, the ETCs fall into the sPCA space much less symmetrically. The strong correlation between the wind speeds (PC1) and WFP (PC3) is evident as most of the tracks are on the same side of the mean of the PC (value 0) for PC1 and PC3 (either calm and small or windy and big). Precipitation (PC2) has a weaker correlation with the winds and WFP, which can be seen in the more even distribution of positive and negative PC3 values on either side of PC2 mean than of PC1 mean. However, more than half of the ETCs ($56\,\%$) are in sectors in which all the three PCs are on the same side of the mean, i.e., in the qualitative binary representation of the sPCA space are either calm, dry, and small ($32\,\%$) or windy, rainy, and big ($24\,\%$).

We use the result of the sPCA to reduce the number of necessary intensity measures in the dataset for comprehensive description of ETC intensity. Despite strong correlations between the winds, WFP, and VO, we retain them separately in the



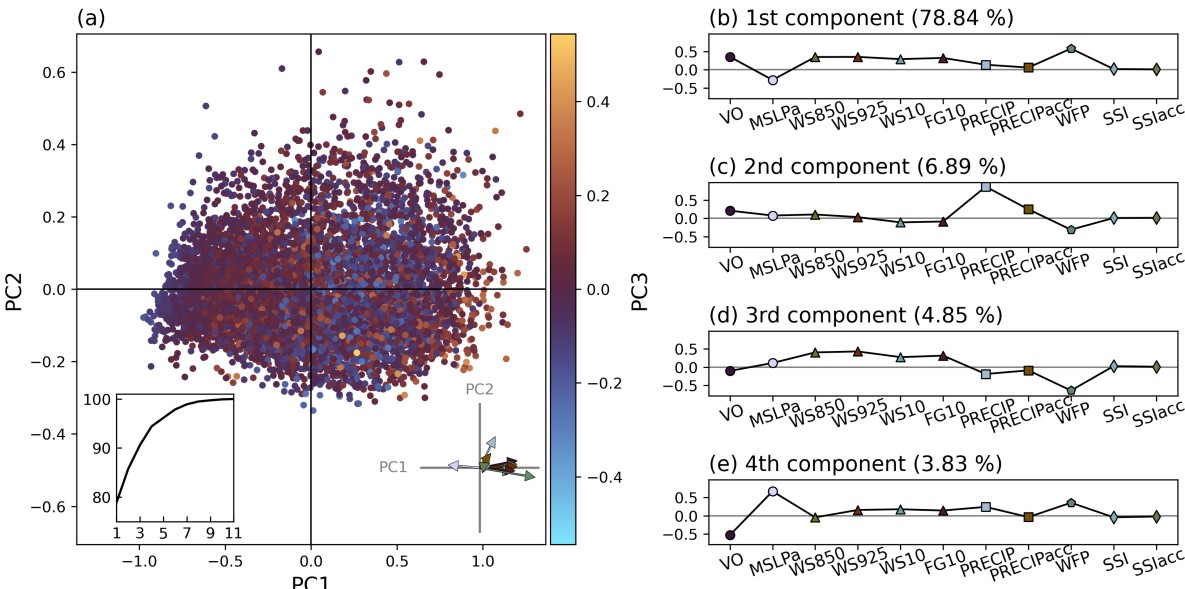

**Figure 4.** (a) ETCs projected onto the PCA space. PC1 and PC2 are the horizontal and vertical axes, respectively, and PC3 is shown in colours. Inset in the bottom left corner shows cumulative proportion of explained variance (percent) as a function of number of PCs. Inset in the bottom right corner shows loadings of PC1 and PC2 as vectors. (b–e) The weights of input measures in the first four PCs. Weights with larger magnitude mean more contribution of the specific intensity measure to the PC. The numbers in parentheses indicate the proportion of total explained variance in a given PC.

reduced set as they appear each on their own in the sPCA. We choose to keep five intensity measures in the final set for comprehensive and non-redundant representation of ETC intensity:

1. WS850: All four wind speed measures are highly correlated with each other and grouped in PC1 of the sPCA. WS850 is chosen because of its link to VO in PC4.

2. PRECIP: From sPCA.

3. WFP: From sPCA.

4. VO: From sPCA. Preferred over MSLPa because of its minor weight in PC1 and PC2.

5. SSI: Although SSI has zero weight in sPCA, it is included since it is weakly correlated with the other intensity measures and has a more non-linear relationship with them. It can therefore be used to separate the feature space better (non-linearly).





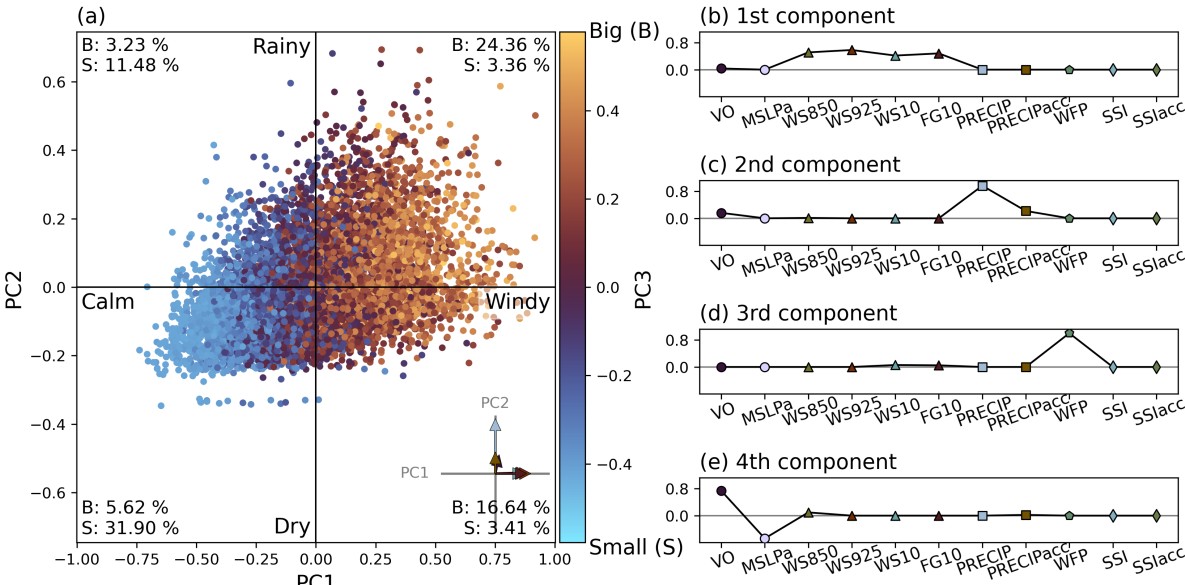

**Figure 5.** (a) ETCs projected onto the sPCA space. PC1 and PC2 are the horizontal and vertical axes, respectively, and PC3 is shown in colours. Inset in the bottom right corner shows loadings of PC1 and PC2 as vectors. The percentages indicate how large proportion of all tracks falls into each sector (on which side of the mean, i.e., negative or positive value of PC). (b–e) Weights of input measures in the four PCs.

## 4.3 Cluster analysis

The cluster analysis was performed using the method described in Sect. 3.3 with the reduced set of intensity measures as input. The number of clusters was chosen to be four. With the GMM method we obtain a cluster centroid and a specific covariance structure for each cluster, i.e., in total four of both. Together the centroids and covariances define a multivariate normal distribution which is used to assign each ETC into the most probable cluster. The four obtained clusters are named 1) HighSSI (proportion of total ETCs in the cluster: $8.57\%$), 2) Intense ($21.46\%$), 3) AvgMST (average, main storm track; $44.42\%$) and 4) Weak ($25.54\%$), based on the average magnitudes of input features and the average geographical locations of ETCs in the clusters which are explained in detail in the following sections.

### 4.3.1 ETC intensity measures

Distributions of all 11 intensity measures for ETCs in each of the clusters are shown in Fig. 6. For each intensity measure, the distributions are significantly different between the four clusters at the $5\%$ level based on a Mann–Whitney U-test (Mann and Whitney, 1947, not shown). The shapes of the intensity measures' distributions in different clusters are largely similar in nature as the full distributions in Fig. 2. For example, in each cluster the distribution of VO is Gaussian-like (Fig. 6a). In terms of both the mean and the median, the average magnitude of VO in the clusters in decreasing order is HighSSI, Intense, AvgMST,





and Weak. Clusters HighSSI and Intense have the widest (i.e., the most variance) VO distributions and cluster AvgMST has a
slightly narrower one, whereas cluster Weak has an even narrower VO distribution. All distributions of WS850 (Fig. 6b) and
WS10 (Fig. 6h) are also Gaussian-like with similar shapes between the clusters and the order of the average magnitudes is the
same as for VO. For MSLPa (Fig. 6f) and WS925 (Fig. 6g) the shapes of HighSSI, Intense, and AvgMST distributions are more
similar between each other than for the previously mentioned intensity measures, but the order of the average magnitudes of the
intensity is the same. FG10 distributions of clusters HighSSI and Intense are less Gaussian-like with almost flat tops (Fig. 6i).
Compared to the other dynamical intensity measures, the distribution of FG10 for cluster HighSSI is also wider relative to the
other clusters. The order of the average magnitudes is, however, the same as for the other dynamical measures.

Like the full distributions in Fig. 2, the distributions of the impact-relevant intensity measures in the clusters are less
Gaussian-like. In the distributions of WFP for the different clusters, the order of average magnitudes is the same as in the
dynamical measures, but there is more overlap between the three most intense clusters (Fig. 6d). Cluster Weak is dominated
by values at the small end of the WFP distribution. For PRECIP, clusters Weak and AvgMST have narrow Gaussian-like distri-
butions, while distributions of clusters Intense and HighSSI are positively skewed (Fig. 6c). The PRECIP distributions largely
overlap, especially between clusters Intense and HighSSI. In fact, PRECIP is the only intensity measure for which the mean
value is the largest for cluster Intense instead of cluster HighSSI. Distributions of PRECIPacc also largely overlap but as op-
posed to PRECIP, all clusters have a long tail on the large end of the distribution (Fig. 6k). Although values of PRECIPacc
are on average the largest in cluster HighSSI, the largest values for single ETCs are found in cluster Intense. SSI distributions
of clusters Weak and AvgMST heavily overlap and comprise most of the smallest SSI values (Fig. 6e). There is little overlap
between the other clusters, as most values in cluster Intense are larger than any value in the two weaker clusters, and almost
all values in cluster HighSSI are larger than any value in the other three clusters. These three distinct ranges of SSI values are
probably an effect of the highly skewed distribution of SSI, and they indicate that SSI creates a lot of separation between the
clusters. For SSIacc there is a clear separation between the mean values in the clusters, but there is much more overlap between
distributions than for SSI.

Because of the relatively direct link between the PCs and the intensity measures, we can infer the distributions of the clusters
in the sPCA space shown in Fig. 5a based on the means and shapes of the distributions of the intensity measures in Fig. 6a–d.
We see that cluster Weak is very compact in the sPCA space since it is always in the smaller end of the intensity and has a very
narrow distribution. Cluster AvgMST is also relatively compact as it always has the second smallest intensity on average and
has less spread than either cluster HighSSI or cluster Intense. The two most intense clusters, HighSSI and Intense, are the least
compact in the sPCA space as they have the highest spread in their distributions. Overlap between the distributions indicates
that there is overlap also in the sPCA space between ETCs belonging to different clusters. This means that the clusters are not
completely distinct in the phase space determined by the intensity measures or the sPCA, but are formed from a continuous
multidimensional distribution. The most overlap is between the distributions of clusters HighSSI and Intense (except in SSI),
which suggests that cluster HighSSI is a subset of cluster Intense that contains the most extreme ETCs on average.





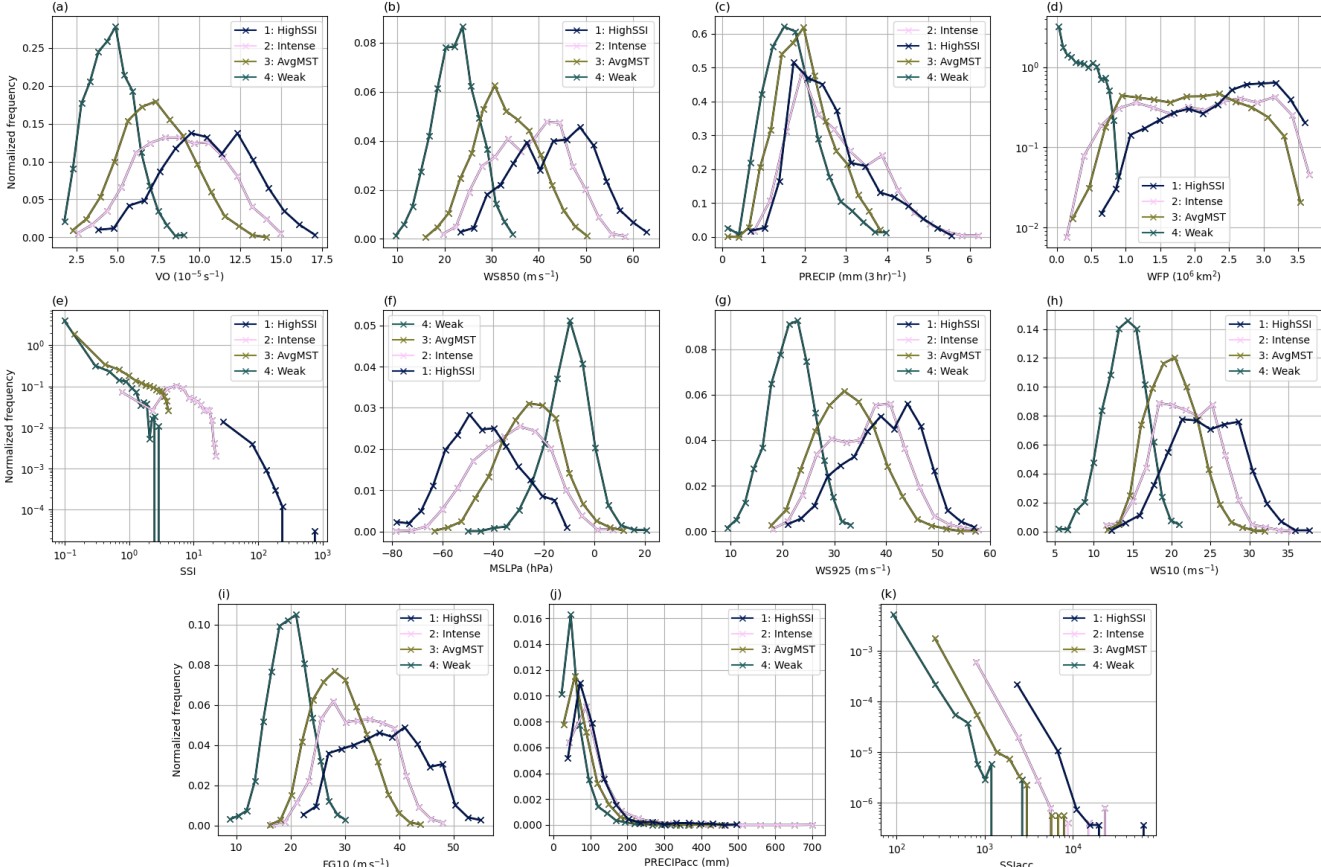

**Figure 6.** Probability densities of the intensity measures in ETC clusters. The legend in each panel is ordered based on the means of the distributions from largest to smallest.

### 4.3.2 ETC characteristics

In addition to the intensity measures, some of which are used to perform the cluster analysis, we compare additional ETC characteristics between the clusters. Figure 7 shows the distributions of latitude of genesis, meridional displacement (latitude

of lysis minus latitude of genesis), latitude of maximum VO, deepening rate (difference in MSLPa 24 hours before and at time of minimum MSLPa), lifetime, and mean displacement speed (averaged over the whole life cycle) of ETCs in the four clusters. The distributions of latitude of genesis overlap considerably between the clusters and they all peak around $40°$ N (Fig. 7a). However, only distributions of HighSSI and Intense are not significantly different at the $5\,\%$ level (not shown). On average, the highest latitude of genesis is in cluster AvgMST ETCs, followed by clusters Weak, HighSSI, and Intense, respectively.

Compared to latitude of genesis, there is more variation between clusters in the meridional displacement of ETCs (Fig. 7b). The largest meridional displacement is on average in cluster HighSSI with a peak around $25°$. Slightly smaller displacements



are found in clusters Intense and AvgMST while cluster Weak has the most negative displacement values (i.e., equatorward displacement) with a peak in the distribution around $-5°$. This causes ETCs in cluster Weak to have the smallest latitude of maximum VO on average, with a peak around $35°$ (Fig. 7c). Although cluster Weak ETCs have on average a higher latitude of genesis than cluster HighSSI and Intense ETCs, the large meridional displacements in the latter two cause them to have on average higher latitudes of maximum VO than the former. Despite the slightly smaller meridional displacement values in cluster AvgMST than in clusters HighSSI and Intense, the highest latitudes of genesis cause ETCs in the former to have on average the highest latitudes of maximum VO, with a peak in the distribution around $60°$ N.

All distributions of deepening rate (Fig. 7d), lifetime (Fig. 7e), and mean speed (Fig. 7f) are skewed to the right. All of them also have the same order of average magnitude between the clusters: 1) HighSSI, 2) Intense, 3) AvgMST, and 4) Weak. This is the same order as in most of the intensity measures in Fig. 6. The order is also the same in meridional displacement, which can possibly be explained by longer-lasting and faster-moving ETCs leading to larger displacements. This link is, however, weaker in cluster Weak, in which many ETCs have equatorward displacement, and the effect of lifetime and mean speed to meridional displacement does not apply in the same way. Relationships between average magnitudes can also be seen between some of the ETC characteristics and intensity measures. For example, higher wind speeds lead to higher displacement speeds and larger deepening rates mean deeper ETCs in terms of MSLPa.

In general, the distributions of ETC characteristics have more overlap between clusters than the intensity measures. As was the situation with the intensity measures, in almost all ETC characteristics clusters HighSSI and Intense have the most similar distributions between each other. Despite this, all distributions are statistically different at a significance level of at least $5\%$ in a Mann–Whitney U-test, except for the latitude of genesis of clusters HighSSI and Intense (not shown). This indicates that the intensity-based cluster analysis is able to identify ETCs which are different in terms of their life cycle characteristics in addition to their intensity.

### 4.3.3 Geographical distribution of ETCs

Figure 8 shows where ETCs in different clusters geographically occur compared to the full climatology. Track densities in Fig. 8a–d are calculated separately for the clusters and normalised by multiplying them with $N_{tot}/N_c$, where $N_c$ is the number of tracks in a cluster and $N_{tot}$ is the total number of tracks. Finally, the track density of the full climatology (Fig. 8e) is subtracted from them to highlight differences between clusters.

In Fig. 8d we see that ETCs in cluster Weak are mostly absent from the main North Atlantic storm track area (the southwest to northeast tilted area of large track densities in the Atlantic in Fig. 8e). They are inversely the most abundant in the Mediterranean basin, which is a local maximum area of track density in the full climatology. Mediterranean cyclones are generally smaller and have shorter life cycles than ETCs in other larger basins (Campins et al., 2011), which may explain the larger number of Weak cyclones in this region as e.g., their WFPs tend to be smaller. The high density of cluster Weak ETCs in the Mediterranean is also consistent with small values of latitude of maximum VO (Fig. 7c) and meridional displacement (Fig. 7b), given the location and orientation of the Mediterranean basin. In addition to the Mediterranean basin, ETCs in cluster Weak comprise most of the tracks in continental Europe (Fig. 8d). Occurrence in this area also explains the smaller WFP values, as near-surface



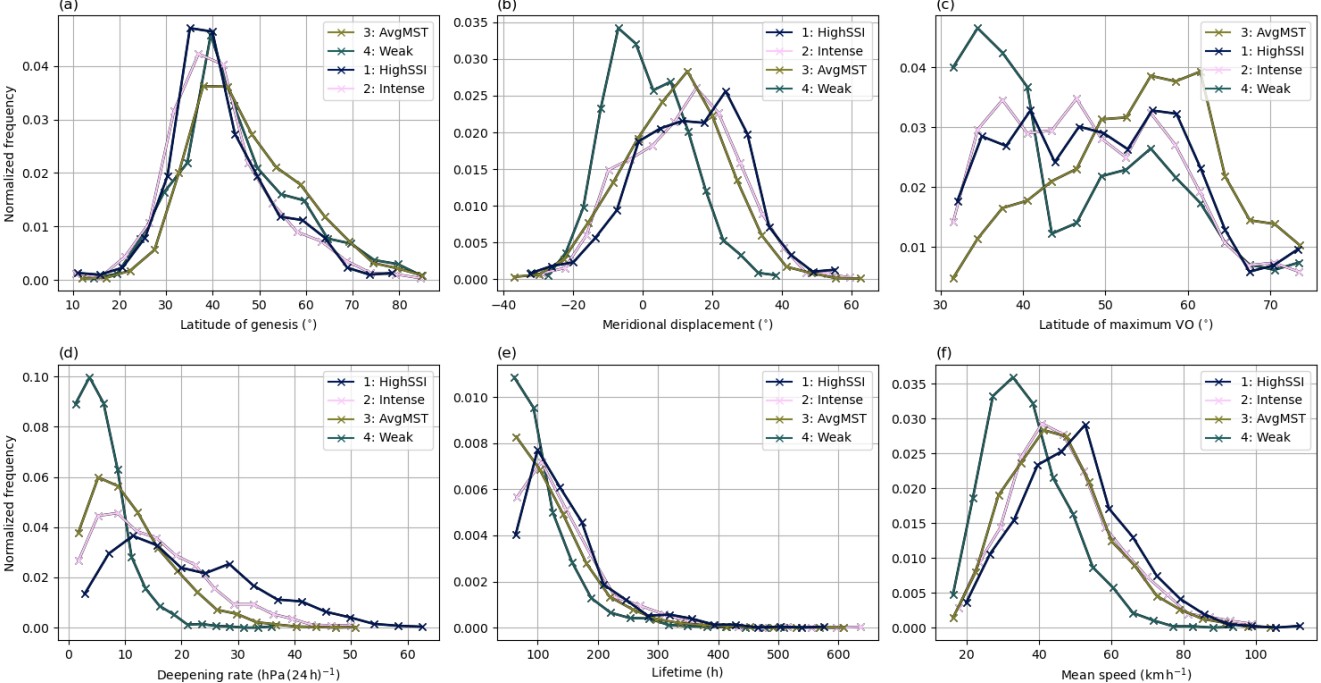

**Figure 7.** Probability densities of the ETC characteristics for the clusters. The legend in each panel is ordered based on the means of the distributions from largest to smallest.

wind speeds and gusts are lower over land than over sea areas (Laurila et al., 2021b). However, it does not explain the small SSI values, which depend on local FG10 values instead of an absolute threshold.

From a qualitative perspective, the occurrence areas of ETCs in cluster AvgMST (Fig. 8c) are a mirror image of the ones in cluster Weak. As the name suggests, ETCs in cluster AvgMST mostly occur along the main storm track, especially in its middle part with a maximum between Greenland and Iceland. In general, ETCs in this cluster occur more in the northern parts of the domain and are largely absent from Europe and south of 45° N. This can be seen also in the northernmost values of genesis latitude (Fig. 7a) and latitude of maximum VO (Fig. 7c). The differences in normalized track density with the full climatology are however smaller in cluster AvgMST than in cluster Weak.

ETCs in cluster Intense have a maximum in track density over the northeastern coast of the United States (Fig. 8b). Most of them occur at the start of the storm track, and elsewhere in the domain, differences are small compared to the full climatology. The location of many of the tracks in this cluster partly explains the large precipitation values in this cluster, as the ETCs occur in the southern parts of the domain and over oceans where surface moisture is abundant.

Due to the small size of cluster HighSSI (8.57 % of all tracks), the normalization of the track density by the total number of tracks in the full climatology affects the distribution greatly, with individual tracks having a larger contribution than in the other clusters. Despite this, the distribution of track density in cluster HighSSI (Fig. 8a) looks like what we would expect based on



its similarity to cluster Intense in terms of intensity and ETC characteristics. As in cluster Intense, areas of high track density are at the start of the storm track and elsewhere differences with the full climatology are small. However, the area of higher track density extends more to the northeast along the storm track (i.e., over the UK and southern Scandinavia) than in cluster Intense, but the values in this area are more discontinuous, which is likely due to the effect of normalization.

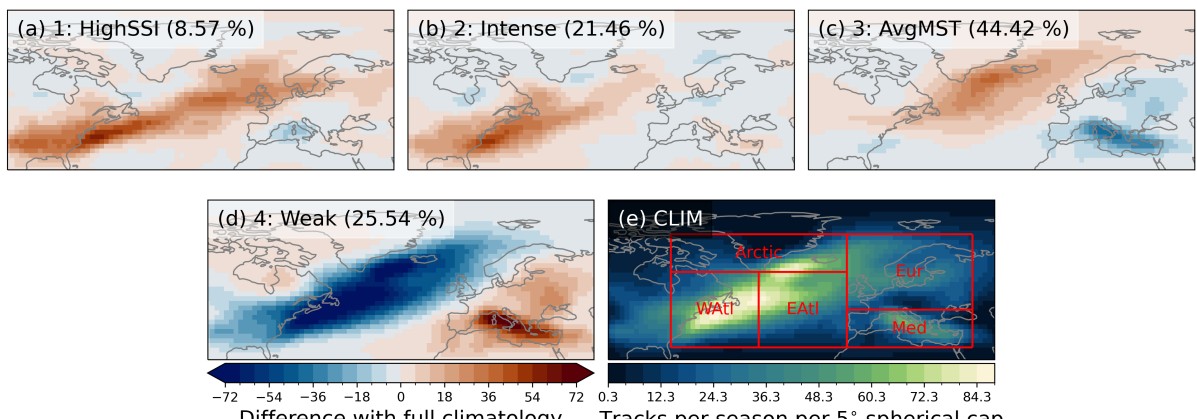

**Figure 8.** (a–d) Anomalies of track density in the clusters compared to (e) the full climatology. Track densities are calculated separately for the clusters, normalised by multiplying them with $N_{tot}/N_c$, where $N_c$ is the number of tracks in a cluster and $N_{tot}$ is the total number of tracks, and the track density of the full climatology is subtracted from them. The percentages in parentheses indicate how large proportion of all tracks is in each cluster.

Figure 9 shows the geographical distribution of ETCs in different clusters from another perspective. It shows the proportion of ETCs in each cluster in the geographical boxes shown in Fig. 8e. Essentially, the results are consistent with the average perspective in Fig. 8. There are mostly cluster Weak ETCs in the Mediterranean and Europe. Over the Atlantic Ocean, most ETCs are in cluster AvgMST, especially in the "Arctic" area where their proportion is more than $60\%$. At the start of the storm track in the Western Atlantic, cluster Intense ETCs comprise a third of all ETCs. However, in this representation we can also see information that is more difficult to see from Fig. 8: almost half of ETCs in Europe are in cluster AvgMST and each area has a similar proportion of HighSSI ETCs (around $10\%$). The latter finding indicates that, given the track density distribution, HighSSI ETCs occur as frequently in each area. This is consistent with the track density distribution in Fig. 8a in which the largest deviations from the climatology of track density occur on a narrow area across the Atlantic. Furthermore, while Fig. 8 may give the impression that almost all ETCs identified in the Mediterranean belong to cluster Weak, Fig. 9 shows that approximately 20 % of ETCs in the Mediterranean belong to cluster Intense, indicating that strong ETCs do develop and are identified in the Mediterranean.





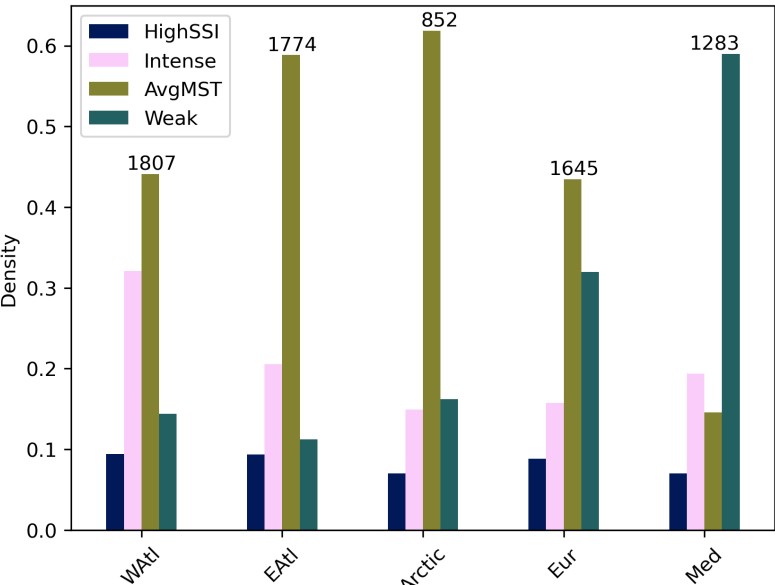

**Figure 9.** Proportion of ETCs in each cluster in the area boxes shown in Fig. 8e. The occurrence area of an ETC is allocated based on its location at time of maximum VO. The values are normalized by the number of ETCs occurring in each box, which is shown by the numbers at the top of the bars. Thus, the four bars for each area sum up to one. Explanations for the abbreviations are: WAtl = Western Atlantic, EAtl = Eastern Atlantic, Eur = Europe, Med = Mediterranean.

#### 4.3.4 Temporal occurrence of ETCs

We investigate the temporal occurrence of the total number of ETCs and the number of ETCs in each cluster with a trend analysis. First, time series of how many ETCs occur in each extended winter are smoothed by taking a 5-year running mean

of them. A Mann–Kendall test (Mann, 1945; Kendall, 1948) is performed on these smoothed time series to detect trends using the Python package pyMannKendall (Hussain and Mahmud, 2019). We find that there is no trend in the total number of ETCs within the study period (not shown). The time series of the number of ETCs in each cluster and the results of the Mann–Kendall test are shown in Fig. 10. There is large interannual variability in the number of ETCs per season, especially in clusters Intense (Fig. 10b) and AvgMST (Fig. 10c), which is a similar result to that found by Laurila et al. (2021a). The slope of the trend

is positive in cluster HighSSI (Fig. 10a) and negative in cluster AvgMST, but these trends are not statistically significant. In contrast, statistically significant (at $1\%$ significance level) increasing and decreasing trends are identified in clusters Intense and Weak (Fig. 10b and d), respectively. To understand why, we computed the trends in all eleven intensity measures for all ETCs (not shown). PRECIP, PRECIPacc, and SSIacc have significantly increasing trends (at the $5\%$ level) while none of the other intensity measures has a trend. The increase in the number of Intense ETCs, which have the highest PRECIP values on

average and the largest PRECIPacc values for single ETCs, is thus consistent with the increasing trend in precipitation in all ETCs. We can therefore say that within our study period, the intensity of ETCs has increased mostly in terms of precipitation.





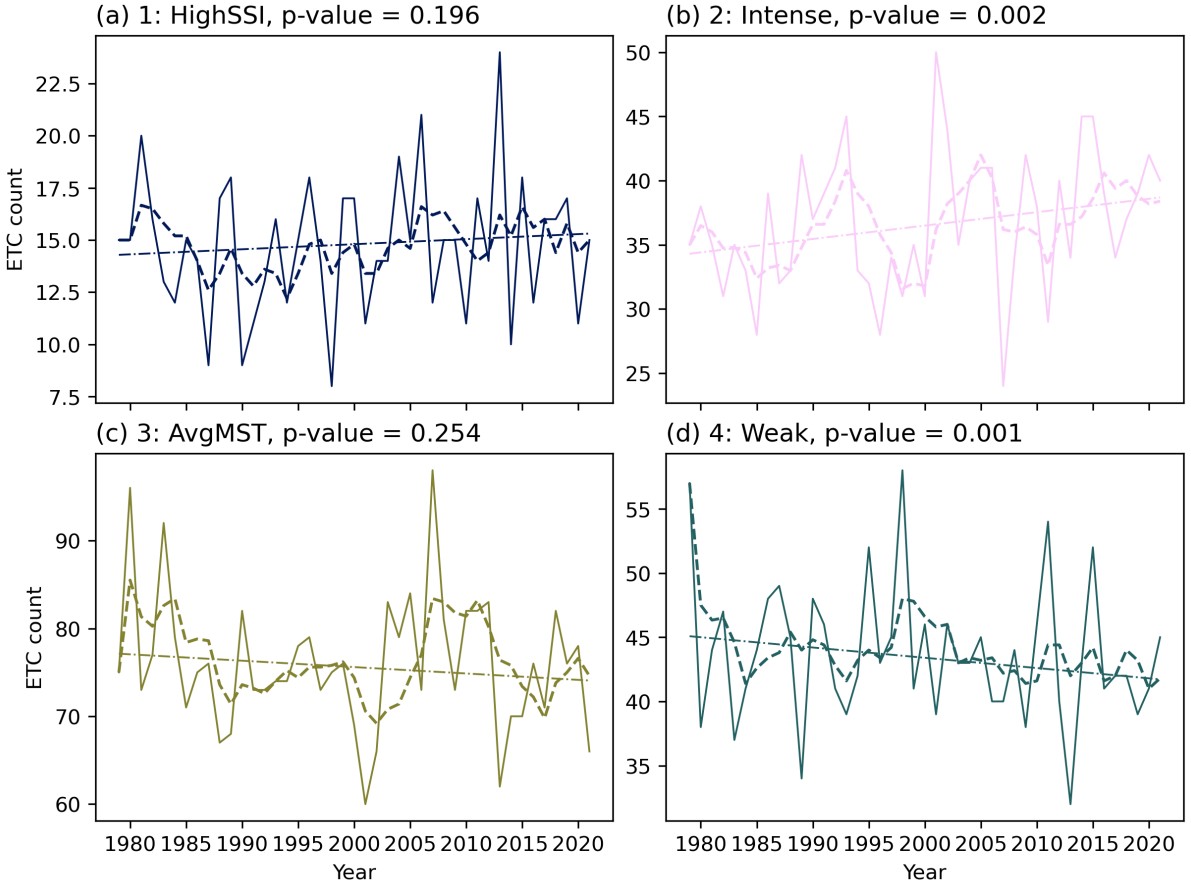

**Figure 10.** Temporal trends of the number of ETCs in each cluster. The solid lines show the number of ETCs in each extended winter season, and the dashed lines show 5-year running means of these. The dash-dotted lines show the slopes of the trend of the 5-year running means from a Mann–Kendall test. Note that the vertical axes have different scales in each panel.

This increasing trend in ETC precipitation is in agreement with Li et al. (2014) who compared a recent warmer period (in average SST) as an analogue of future climate change to an earlier base period and found that while precipitation was larger in the warmer period than the base period, there was no consistent change in e.g., vorticity or wind speed. The significant increase in SSIacc, without an increase in FG10, can be explained by an increase in extreme wind gust values in North Atlantic ETCs from 1979 to 2021 found by Karwat et al. (2022).

## 4.4 Case study ETCs

In addition to the statistical approach, we investigate the clusters in terms of individual ETCs. We select named ETCs which mainly affected Europe and occur either in the XWS storm catalogue (Roberts et al., 2014) or a list of strong storms in Finland maintained by the Finnish Meteorological Institute (FMI, 2023), or are well known intense Mediterranean cyclones.



The selected ETCs are listed in Table 2. Figure 11 shows these storms projected onto the first two PCs of the sPCA space and coloured by either their cluster (Fig. 11a) or the value of PC3 of the sPCA (Fig. 11b).

Figure 11a shows that cluster HighSSI is disproportionately represented in the set of named storms. Out of the 21 storms, 17 (81 %) are assigned to cluster HighSSI despite it consisting of less than 10 % of all ETC tracks in the dataset. The four storms which do not belong to cluster HighSSI have the smallest SSI values of the 21 case studies, which cannot be seen in the PCs in Fig. 5. They are also among the five storms with the smallest PC4 values, which means they have low VO and/or high MSLPa values. In Fig. 11b we see that most of the storms fall close to the mean value of PC2, meaning that they have near-average precipitation. All storms which affected Europe are relatively close to one another in the sPCA (e.g., Anatol, Kyrill, and Ulli), except for Vivian, Xynthia, and Apollo which had more precipitation and/or lower wind speeds. Although the cluster HighSSI is overwhelmingly the most common cluster among the named storms, only nine (43 %) of the storms have a positive value in all four PCs. Therefore, we can conclude that the cluster analysis can also identify impactful storms which do not have high values in all PCs (e.g., Christian/St. Jude has negative PC2 and PC3 values, and a small PC4 value but still is a HighSSI storm). This demonstrates the need to use more than one measure to quantify ETC intensity.

At the same time, there is only one storm, medicane Apollo, which has a negative PC1 value, i.e., smaller than average wind speed, and a PC3 close to the negative end, i.e., small WFP. Apollo, however, has the sixth largest PC2 value of the 21 case study ETCs, which is to be expected since in Apollo most of the damage was caused by precipitation rather than wind. Apollo belongs to cluster Weak, since the cluster analysis discriminates intensity more based on wind than precipitation (cf. e.g., Fig. 6b and Fig. 6c). In the case of Apollo, however, this may be due to the underestimation of the medicane's intensity in ERA5 (Pantillon et al., 2024).

There are a couple of reasons why most of the selected storms are more extreme in terms of wind (PC1) than precipitation (PC2). Firstly, the majority of them come from the XWS catalogue (Roberts et al., 2014), in which the storms are selected using wind-based diagnostics. Secondly, ETCs with the highest PC2 values, and thus the most precipitation, occur mostly over the ocean, where their effects are not felt and storms do not get named. Moreover, ETCs which do have high precipitation values over land areas occur mostly over North America (not shown). This is demonstrated by the three North American storms in the top right corner of the sPCA space: Ex-hurricanes Noel and Wilma, and the northeaster 1993 Storm of the Century. However, some of the storms (e.g., Xynthia, Vivian, and 1993 Storm of the Century) had compound impacts with heavy rainfall and strong winds associated with them.

This analysis indicates that SSI is important in determining the clusters of ETCs with similar PC values. Based on the case study storms, ETCs with large SSI values are allocated to cluster HighSSI even though they have less than average values in other intensity measures. However, cluster analysis is a statistical tool and we cannot say that all ETCs in cluster HighSSI are significantly impactful or that all impactful ETCs belong to cluster HighSSI. This is demonstrated by the two case study storms, which belong to clusters AvgMST and Weak.



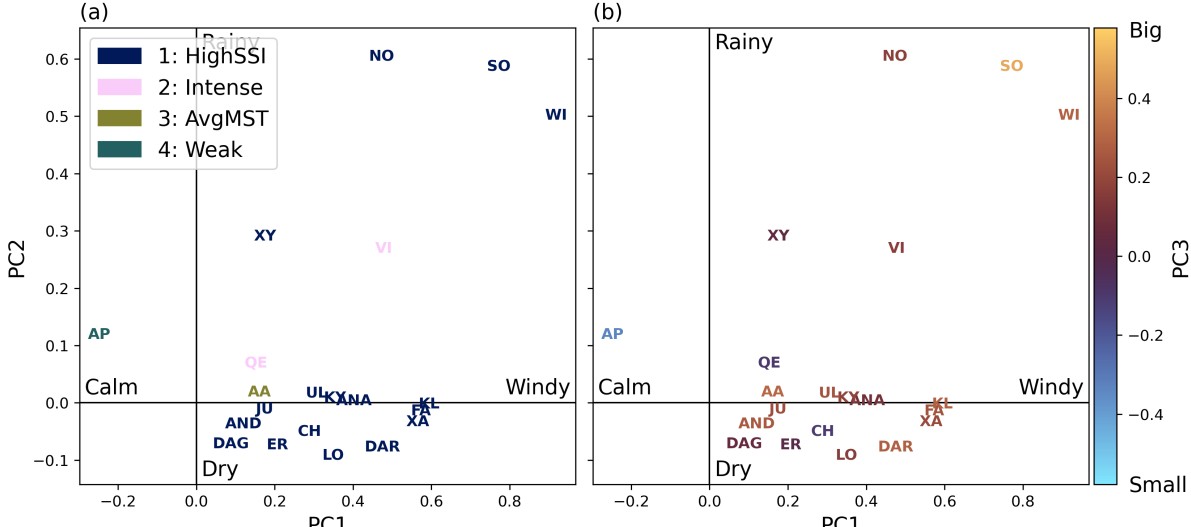

**Figure 11.** Case study storms in the sPCA space, coloured by (a) their predicted cluster and (b) their PC3 value. See Table 2 for abbreviations of storm names.

## 5 Discussion

Classes of ETCs with similar distinctions have been found with cluster analysis methods in previous studies. Blender et al.
(1997), who used *k*-means clustering on Lagrangian North Atlantic ETC tracks from ECMWF operational analyses, found three clusters of track orientations: stationary, northeastward, and zonal. Qualitatively, our findings match theirs well. Their stationary ETCs occurred mostly in the Mediterranean, Greenland, and Northern Canada and, as the name suggests, had small propagation speeds. This is similar to our cluster Weak. Likewise, their northeastward ETCs are similar to our clusters HighSSI and Intense, with large meridional displacements and propagation speeds. While their zonal ETCs match our cluster AvgMST
in terms of the more moderate meridional movement, their zonal tracks are not concentrated at the end of the storm track in Northeastern Atlantic like our AvgMST ETCs are. Instead, their northeastward and zonal clusters and the pressure patterns associated with them are reminiscent of the difference between North Atlantic ETC track regimes during positive and negative North Atlantic Oscillation (NAO) phases (Rogers, 1997; Hodges, 2008). In addition to the similarity in track orientations, ETCs in their clusters had similar average intensities as ours. Their stationary ETCs had on average the weakest 1000 hPa geopotential
height gradient (comparable to low-level winds) and the highest 1000 hPa geopotential height (comparable to MSLP), while northeastward ETCs had the strongest 1000 hPa geopotential height gradient and the lowest 1000 hPa geopotential height values.

Similarly, Gaffney et al. (2007) performed cluster analysis on Lagrangian North Atlantic ETC tracks between 1958 and 2002. They used the National Centers for Environmental Prediction (NCEP) NCAR reanalysis data and regression mixture
models, which are probabilistic methods like GMM. Like Blender et al. (1997), they found clusters with northeastward and



| Name | Abbrev. | Occurrence | Affected area | Reference |
|---|---|---|---|---|
| Aapeli/Alfrida | AA | Jan. 2019 | Northern Europe | ECMWF (2021) |
| Anatol | ANA | Dec. 1999 | Northern Europe | Ulbrich et al. (2001) |
| Andrea (secondary) | AND | Jan. 2012 | Mediterranean | Kouroutzoglou et al. (2013) |
| Apollo/Nearchus | AP | Oct. 2021 | Mediterranean | Menna et al. (2023) |
| Christian/St. Jude | CH | Oct. 2013 | Northwestern Europe | Hewson et al. (2014) |
| Dagmar/Patrick/Tapani | DAG | Dec. 2011 | Northern Europe | Weijenborg and Spengler (2020) |
| Daria/Burns' Day Storm | DAR | Jan. 1990 | Northwestern Europe | McCallum (1990) |
| Erwin/Gudrun | ER | Jan. 2005 | Northwestern Europe | Suursaar et al. (2006), Baker (2009) |
| Fabien | FA | Dec. 2019 | Mediterranean | Stojanovic et al. (2021) |
| Julia | JU | Feb. 2012 | Mediterranean | Metheniti (2012) |
| Klaus | KL | Jan. 2009 | Southwestern Europe | Liberato et al. (2011), Bertotti et al. (2012) |
| Kyrill | KY | Jan. 2007 | Western Europe | Fink et al. (2009) |
| Lothar | LO | Dec. 1999 | Western Europe | Ulbrich et al. (2001), Wernli et al. (2002) |
| Ex-Hurricane Noel | NO | Nov. 2007 | Eastern North America | Brennan et al. (2009) |
| Qendresa | QE | Nov. 2014 | Central Mediterranean | Coll-Hidalgo et al. (2022) |
| 1993 Storm of the Century | SO | Mar. 1993 | Eastern North America | Huo et al. (1995) |
| Ulli | UL | Jan. 2012 | Northwestern Europe | Fox et al. (2012), Smart and Browning (2014) |
| Vivian | VI | Feb. 1990 | Western Europe | Schüepp et al. (1994) |
| Ex-hurricane Wilma | WI | Oct. 2005 | Eastern North America | Pasch et al. (2006) |
| Xaver | XA | Dec. 2013 | Northern Europe | Hewson et al. (2014) |
| Xynthia | XY | Feb. 2010 | Western Europe | Liberato et al. (2013), Ludwig et al. (2014) |

**Table 2.** The named case study ETCs included in Fig. 11. The columns are the name of the ETC, abbreviation used in Fig. 11, time of occurrence, the area the ETC mostly affected, and a reference study for the ETC, respectively.

zonal track orientations. They, however, identified also a cluster with northward track orientations, which could not be likened to any of the clusters of Blender et al. (1997), and they did not find a cluster of stationary ETC tracks. Their northward-oriented tracks were mostly found near the eastern coast of North America, which means it could contain many of the same tracks as in our clusters HighSSI or Intense (e.g., post-tropical cyclones). In terms of MSLP, their northward-oriented ETCs were the most
590 intense but the northeastward-oriented ETCs were faster-moving, while they found no significant differences in ETC lifetime between any of the clusters. Gaffney et al. (2007) also performed analysis on the relationship between the ETC clusters and the large-scale background flow. They found that northeastward-oriented ETC tracks occur in positive NAO phases and zonally oriented ETC occur in negative NAO phases, which is qualitatively consistent with the results obtained by Blender et al. (1997).

Leckebusch et al. (2008b) used a different approach to relate large-scale flow patterns to different types of ETCs. First, they
performed PCA on the 1000 hPa geopotential height field from the NCEP–NCAR reanalysis dataset to reduce the amount of data. From the PCA, they identified six large-scale flow patterns over Europe, which they then used to classify winter storm



situations with *k*-means clustering. They identified 55 clusters of pressure patterns, of which four were classified as primary storm clusters and nine as secondary storm clusters. These storm clusters were associated with more extreme ETCs as 72 % of 46 important European winter storms occurred during the four primary storm clusters, while the overall relative frequency of occurrence of these four clusters was only 5 %. This result is reminiscent of our finding that the majority of important storms are found in cluster HighSSI despite its small proportion of all ETCs.

Others have also previously used phase spaces consisting of different variables to categorize ETCs. Many have subjectively divided phase spaces into various parts and analyzed ETCs in each part of the phase space separately. Graf et al. (2017) performed PCA on 30 ETC precursors to classify Northern Hemispheric cyclogenesis events with ECMWF's ERA-Interim reanalysis data. Although they found no obvious clusters in the continuous phase space determined by the genesis events, using the first two components of the PCA they were able to formulate five ETC classes: $A_{moist}$, $A_{dry}$, $B_{moist}$, $B_{dry}$, and M. The first PC determined whether an ETC was characterized by strong (moist) or weak (dry) moist processes in its genesis, and the second PC split the genesis events based on their likeness to the ETC life cycle classes of Petterssen and Smebye (1971), A and B. Their fifth class, M for "middle class", consisted of ETCs which fell close to the mean of the phase space defined by the first two PCs. Their PCA classification was robust to the number of input features, with five precursors producing similar results as the initial 30. They also determined that a majority (67 %) of investigated well-known ETCs belonged to a single class, namely $A_{moist}$. All four case study ETCs that are shared between their and our studies (Klaus, Kyrill, Lothar, and Xynthia) belonged to the class $A_{moist}$, while in our investigation they were all found in cluster HighSSI. While they note that their analysis cannot be used to directly attribute cyclogenesis events to specific cyclone evolution, this is an interesting result.

Besson et al. (2021) investigated dry-dynamic forcing of Northern Hemispheric ETCs by studying their Eady growth rate and upper-level induced quasi-geostrophic ascent calculated from ERA-Interim reanalysis. They defined four categories of ETC forcing by selecting values at the extreme corners of a two-dimensional phase space determined by the two variables. They found that these four categories of ETC forcing occur in different geographical areas, and lead to ETCs which differ in their deepening rates and have differences in their upper-level structure. Similarities between their categories and our clusters can be seen in the link between the ETC deepening rates and occurrence areas of the four categories of ETC forcing. For example, a combination of the two forcing mechanisms leading to large deepening rates are found mostly at the start of the North Atlantic storm track (cf. clusters HighSSI and Intense) while combinations leading to the smallest deepening rates occur more at the southern parts of the North Atlantic, towards the end of the storm track (cf. cluster Weak). Deepening rates in between these extremes are associated with a combination of forcing mechanisms which occurs mostly in the northern parts of the North Atlantic with a maximum density between Greenland and Iceland (cf. cluster AvgMST). Another similar analysis was done by Binder et al. (2016) and Binder et al. (2023), who determined three categories of ETC intensification for Northern Hemisphere ETCs from a phase space of ETC deepening rate and low-level warm conveyor belt air mass using ERA-Interim reanalysis data and CESM-LE model data. They found differences in the structure and occurrence areas of ETCs between the three categories.

While these types of analyses are suitable for studying the precursors and forcing mechanisms of ETCs, we demonstrate that classification of ETCs based on their intensity benefits from an added level of objectivity via the cluster analysis. This



can be seen in the overlap between the intensity measure distributions for different clusters in Fig. 6. In fact, a possible course of future study is the identification of the variability in the ETC precursors and forcing mechanisms within our clusters. We believe this form of analysis would offer a new perspective to the classification of ETC life cycles and possibly improve the predictability of ETC intensity.

A limitation of our study is the use of only one data source (ERA5) and one ETC tracking algorithm (TRACK). Firstly, the representation of ETCs has been found to vary between reanalysis datasets (Hodges et al., 2011; Wang et al., 2016). Secondly, earlier studies have shown that ETC climatologies may differ in distributions and trends due to sensitivity to the tracking algorithm because using different variables and thresholds leads to identifying different categories of ETCs (Raible et al., 2008; Neu et al., 2013; Flaounas et al., 2023). Another limitation of our study is that the criteria used for the ETC tracking are optimized for the North Atlantic. Therefore, e.g., some Mediterranean ETCs are excluded from our set of tracks since they can be more stationary, have shorter lifetimes, and have their maximum vorticity within the first 24 hours. Flaounas et al. (2023) compared 10 ETC tracking algorithms, one of which was TRACK. In their comparison TRACK produced the longest lifetimes and largest propagation speeds for ETCs in the Mediterranean. Despite these limitations, our study brings valuable new knowledge on which intensity measures should be used to comprehensively and non-redundantly quantify the intensity of ETCs.

## 6 Conclusions

We created a dataset of ETC intensity measures for 43 extended winters of North Atlantic and European ETC tracks and performed sPCA to identify the measures which explain most of the variability in the dataset. Using the results of the sPCA and correlations between the intensity measures, we determined five measures for comprehensive and non-redundant representation of ETC intensity. We used these five measures as input in a cluster analysis using GMM to create classes of ETCs based on their multidimensional intensity. The cluster analysis produced the best results with four clusters.

Our analysis shows that while there is strong correlation between different dynamical ETC intensity measures, there is a much weaker link between the dynamical intensity and impact-relevant measures. A correlation of similar strength between wind speed and precipitation was found previously by Pfahl and Sprenger (2016) who determined a correlation of coefficient of 0.36. Therefore, when using ETC intensity as a broad term, i.e., including the impacts as well as the meteorological intensity in the definition, we need to consider the non-linear and weakly correlated relationship between the two, and use more than one or two measures to describe the intensity. Our analysis determined five measures which comprehensively describe ETC intensity: 850 hPa relative vorticity, 850 hPa wind speed, wind footprint, precipitation, and a storm severity index. We recommend studies which aim to quantify future changes in ETC intensity to consider these five variables.

Using these five measures as input to a cluster analysis performed with GMM, we found four clusters in which ETCs were significantly different in terms of their intensity. One cluster had on average weak ETCs (cluster Weak), another one average ones (cluster AvgMST), and two clusters more intense ones (clusters Intense and HighSSI), with the most overlap in the distributions of the intensity measures in the latter two. This overlap, the small size of cluster HighSSI and the large SSI



values in its ETCs indicate that cluster HighSSI is a subset of cluster Intense which contains the most extreme and potentially impactful ETCs. Of all intensity measures, there is the least overlap between clusters in SSI. This, along with the fact that a much larger proportion of named impactful storms than all ETCs belong to cluster HighSSI, suggests that SSI is useful in identifying impactful ETCs, which is in agreement with previous studies (Klawa and Ulbrich, 2003; Leckebusch et al., 2007, 2008a; Donat et al., 2011). The clusters were different in terms of their characteristics and geographical location of

occurrence as well. The weaker ETCs occur more frequently in Europe and the Mediterranean basin than over the Atlantic Ocean, but HighSSI ETCs are almost as frequent everywhere, given the distribution of ETC occurrence in the domain. The number of ETCs in cluster Intense increased from 1979 to 2022 while the number of ETCs in cluster Weak decreased.

An investigation of well-known case study ETCs showed that proportionally, the majority of impactful storms belong to cluster HighSSI. This highlights the ability of the cluster analysis to identify intense storms. However, we cannot say that

all ETCs in cluster HighSSI are impactful or damaging. In addition to the statistical nature of the cluster analysis, this is due to the fact that apart from SSI, which is based on local climatological values, the impact-relevant intensity measures do not discriminate between land and sea areas. Many ETCs with large WFPs and high precipitation values occur mostly over the ocean, which inevitably leads to some non-impactful ETCs being classified as Intense or HighSSI. On the other hand, the analysis showed that SSI, which does not by definition have larger values over sea than land, is an important measure

in determining the cluster of ETCs whose other intensity measures have close to average values. However, SSI alone cannot be used to identify impactful storms since it is relevant for impacts due to wind only, while other factors in ETCs such as precipitation can also cause significant damage. This further emphasizes the fact that multiple intensity measures should be used to quantify ETC intensity.

Our objective classification of ETCs based on their intensity offers a new perspective to the multitude of ETC classifications

reviewed in Catto (2016). Our classification is performed using variables which are available or easy to compute from both model and reanalysis data. Building on the work of Bengtsson et al. (2009) and Champion et al. (2011), who studied the intensity and extreme weather from ETCs in future climates, utilizing this kind of classification in climate projection studies could give insight into how different kinds of ETCs respond to climate change. In addition to climate applications, quantifying the intensity of ETCs is also important in terms of numerical weather prediction. This is especially true in the current age,

when ensemble prediction systems are key to produce probabilistic forecasts and vast amounts of data. Being able to identify ETCs in each ensemble member and compute their intensity allows for an accurate estimate of the uncertainty in how strong, and potentially impactful a specific ETC will be. Furthermore, this allows a vast amount of information to be condensed to a level that is manageable for operational forecasters, who often are working under time pressure.

*Code and data availability.* ERA5 reanalysis data was downloaded from the Copernicus Climate Change Service (Hersbach et al., 2017).

All processed data and Python code are available in a Zenodo repository (Cornér et al., 2024).



*Author contributions.* JC, CB, and VAS all contributed to the design of the study. JC performed most of the data analysis and visualization. JC, CB, and VAS all contributed to the interpretation of the results. BD contributed to the definition and analysis of the storm severity index. FP and BD contributed to the analysis over the Mediterranean region. JC wrote sections 2, 4, 5, and 6. CB wrote section 3. VAS and FP wrote section 1. All authors reviewed and edited the manuscript. VAS secured funding for the study.

*Competing interests.* The authors declare that they have no conflict of interest.

*Acknowledgements.* We wish to thank Kevin Hodges for providing the cyclone tracking software TRACK and support in its set-up and running. We acknowledge CSC – IT Centre for Science, Finland, for computational resources and ECMWF for producing ERA5 reanalysis. This research was supported by the Academy of Finland (grant no. 338615) and is a contribution to the COST Action CA19109 "MedCyclones: European Network for Mediterranean Cyclones in weather and climate". JC was partly funded by the University of Helsinki Doctoral School. 705 BD was funded by Région Occitanie and Météo-France through project PREVIMED. This study uses scientific colour maps (Crameri, 2023) to prevent visual distortion of the data and exclusion of readers with colour vision deficiencies.



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
