# Peer review of "Classification of North Atlantic and European extratropical cyclones using multiple measures of intensity"

_EGUsphere, 2024_

## Referee Comment (RC1)

Review of "Classification of North Atlantic and European extratropical cyclones using multiple measures of intensity" by J. Cornér et al.

General comments:

This study uses ERA5 reanalyses for 43 extended winters to classify North Atlantic and European extratropical cyclones according to their intensity, based on multiple dynamical and impact-related measures. They perform a principle component analysis and a cluster analysis to find four clusters of extratropical cyclones that differ in terms of intensity, geographical location and characteristics. Cluster HighSSI is associated with the highest mean values in many intensity measures (e.g., low-level vorticity and wind speed), followed by cluster Intense. Despite a relatively small occurrence frequency (<10% of all cyclones), HighSSI contains a large fraction of well-known impactful storms. Most cyclones belong to cluster AvgMST (~44%). They have average intensities and occur preferentially over the main North Atlantic storm track. Cyclones in cluster Weak occur in about 25% of all cases, have relatively weak intensities and mainly occur in the Mediterranean and over Europe.

Overall, the manuscript is interesting to read, the storyline is clear, the methods are appropriate, and the general conclusions are sound. Below are a few minor comments and suggestions.

Specific comments:

1) The paper is rather long. This is not a criticism in itself, as the paper is well-written and interesting. However, maybe you can consider shortening the description of the datasets and methods a bit, such that the reader does not have to wait until page 11 for the results. For instance, although nicely written, it might not be necessary to mention all advantages and disadvantages of ERA5, which is a well-known and widely used reanalysis dataset. And in the result section you might consider moving Fig. 9 and its description to the supplement, to shorten the manuscript a bit (but I would also be fine if you left it in the main part).

2) Abstract and conclusions: Most of the abstract describes your method, but it would be nice if you could add some sentences about your results (e.g., the percentage of cyclones in each of the four clusters and the differences between the clusters in intensity, geographical location and characteristics, etc.). Similarly in the conclusions. As the paper is quite long, a summary of the main results would be helpful.

3) Fig. 8a-d: Please add the units to the colorbar.

4) Fig. 10: Do I understand correctly that the positive trend in the number of intense cyclones and the negative trend in the number of weak cyclones indicates that the cyclones moved from the weak to the intense cluster because of the increase in cyclone-related precipitation?

5) Section 4: To define the clusters, you use four wind-related measures (WS850, VO, WFP and SSI), but only one measure for precipitation. Is this the reason why the cluster analysis discriminates intensity more based on wind than precipitation (see for instance Fig. 6b, c), and that cyclones with heavy precipitation but weak winds like Apollo belong to cluster weak rather than intense? Is this a weakness of the method?

6) Section 4: I find it interesting that many of the well-known storms have below-average precipitation (Fig. 11). You say that this is because you picked many of the storms from the XWS catalogue that uses wind-based measures. I know the paper is already rather long, but maybe you can consider including a few more European cyclones that led to heavy precipitation events. It would be interesting to see whether they are located close to Apollo or rather somewhere else in the phase space.

7) Section 4, line 558: "Moreover, ETCs which do have high precipitation values over land areas occur mostly over North America (not shown)." Where do you get this statement from, is it from your dataset? Over Europe ETCs can also produce substantial amounts of precipitation.

Typos and wording:

8) Line 79: "Section 4 contains .. and Section 6 concludes …"
Section 5 is not mentioned.

9) Line 118: "small scale" – maybe better: "small-scale"

10) Lines 135: "beginning from" should be "beginning at"

11) Line 148: Should it be "obtain" instead of "contain"?

12) Line 151: "… and are available as in the reanalysis." The wording is a bit cumbersome, maybe you can reformulate the sentence.

13) Fig. 1: The abbreviation "ONDJFM" has not been introduced. It would fit in the last sentence of Section 2.1.

14) Line 167: remove empty space between double parentheses.

15) Lines 210, 214, 215: "in (each) grid point" should be "at (each) grid point"

16) Line 211: "… whereas Leckebusch et al. (2008a) …" This part of the sentence is difficult to understand, consider reformulating it. Is it actually necessary?

17) Line 234: "For PRECIP, the chosen time is 12 hours before the time of maximum VO, for the same reason." Can you be a bit more specific and briefly motivate why you investigate PRECIP 12h before the time of maximum VO, i.e., at a different time than the other variables? (It does make sense to me, as the peak in precipitation often occurs before the peak in VO, and I guess you suggest this when you write "for the same reason", but the sentence is a bit vague.)

18) Line 296: "all 11 of the intensity measures" – I would write "all 11 intensity measures"

19) Table 1: The description of the column "Distance" could be more precise in the caption.

20) Caption Fig. 2: "times in the tracks" – I would write "times along the tracks"

21) Caption Fig. 3: "An absolute value of Pearson's r is shown for correlations involving MSLPa to aid comparison with other coefficients." I don't know what you are referring to (I don't see anything that is different for MSLPa compared to the other measures).

22) Line 474: "Atlantic" – I would write "North Atlantic"

23) Line 489: Do you mean the southeastern instead of the northeastern coast?

24) Line 490: "… storm track, and elsewhere …" I would split the sentence into two: "… storm track. Elsewhere …"

25) Line 496: consider reformulating the sentence to "As in cluster Intense, the highest track densities occur at the start of the storm track." The second part of sentence I would leave away, as the values are increased everywhere in the main storm track and not only at the beginning.

26) Line 499: What do you mean by "discontinuous"? Consider reformulating the sentence.

27) Caption Fig. 6: "The percentages in parentheses indicate how large proportion of all tracks is in each cluster." Better: "… indicate the proportion of all tracks in each cluster"

28) Line 502: "over Europe"

29) Line 586: "likened" – Typo: "linked"

30) Line 591: "performed analysis" should be "performed an analysis" or "performed analyses"

31) Line 608: Maybe you can briefly mention what the Petterssen and Smebye types A and B are.

32) Line 665: "correlation of coefficient" should be "correlation coefficient"

33) Lines 662 and 669: I would change "had" to "has" and "were" to "are" to consistently use present tense.

34) Line 666: "This, along with the fact that a much larger proportion of named impactful storms than all ETCs belong to cluster HighSSI, …" Complicated wording, consider reformulating it.

35) Line 671: "… but HighSSI ETCs are almost as frequent everywhere, given the distribution of ETC occurrence in the domain." This part of the sentence is a bit difficult to understand, consider reformulating it.

---

## Referee Comment (RC2)

Classification of North Atlantic and European extratropical cyclones using multiple measures of intensity
NHESS-2024-1749

This paper aims to produce cyclones classes which capture various aspects of cyclone intensity better than metrics that use a single diagnostic. Overall, the paper describes the motivation and methods used well. There are however several parts of the paper that require clarification, particularly the clustering method, and more work is required to illustrate how future studies can implement the recommendations made. These points are described below in more detail and should be addressed before the paper is suitable for publication in NHESS.

General comment
1. My main concern is that the 4 clusters identified by the authors are not straightforward to implement by others in future studies. The authors recommend that 5 variables are considered, but how to combine these to identify different classes of cyclone in future studies is not clear. Will subsequent studies need to repeat the Gaussian mixture model with their own tracked cyclone data to identify the 4 clusters? Also, the clusters are identified from the sPCA figure, so that step would also need to be repeated I believe. I would like the authors to provide a more step-by-step guide to how the intensity measures should be combined to 'comprehensively and non-redundantly quantify the intensity of ETCs' (lines 645 and 659). If the cyclone clusters are to be used to see how different kinds of ETCs respond to climate change (line 688) simpler instructions are needed on how to create them. On a related point, on line 692 the authors state that their method 'allows a vast amount of information to be condensed to a level that is manageable for operational forecasters', how would the forecasters use the information? Is it envisaged that they would be provided with the 'intensity' of a cyclone based on a score from each of the clusters, or do the authors have something else in mind?

Specific comments
1. Line 11: What do the authors mean by 'impactful storms'?
2. Line 35: Vorticity is a noisy field typically including both mesoscale and synoptic scale features. Are the authors referring to a filtered vorticity field when they say that vorticity metrics describe the synoptic scale dynamics of ETCs?
3. Line 46 and 54: What is meant by 'concise' metrics? If the aim is to produce a concise metric, this should be defined.
4. Line88: What is $T_1639$?
5. Line 102: What is the consequence of ignoring the biases in ERA5 data? Are you results sensitive to these biases?
6. Line 125: What are the time steps referred to here?
7. Line 151: '… available as is in the reanalysis.' This sentence does not make sense to me.
8. Line 163 and 177: What is the consequence of a mismatch or even no match between the location of the vorticity maxima and mslp minima? Is the latter a consequence of the fact that vorticity can capture the early stages of cyclone development before a closed isobar is identified in mslp?
9. Line177, 330: Here and elsewhere the authors refer to ERA5 wind gusts. It would be useful to have a brief explanation of this diagnostic quantity and how it is derived? Why is it underestimated in some areas (line 206)?
10. Table 1: What is the difference between accumulated and time-integrated?
11. Line 252: Do the authors use the 'type' of correlation, i.e. linear or non-linear later in the analysis or interpretation of their results? I may have missed this. Does the fact that the MI correlations are higher for SSI than the Pearson correlation imply that they are non-linearly related to the windspeed for example?
12. Line 255: Why is it important to know that the method is 'heavily' used? This does not imply that it is the most appropriate method for this study.
13. Lines 281: What is the silhouette score?
14. Figure 2 caption: The caption should refer to table 1 for details of the intensity measures.

15. Line 365: Why do the SSI measures have no weight?  What is the interpretation of this result?
16. Line 395: Here the 4 clusters and their names are introduced.  Are the clusters identified from the sPCA figure (fig 5)? Or have I misunderstood the methodology here? Also, since these names are used frequently in the remainder of the paper, I would suggest using bullet points so that they stand out in the text.
17. Figure 5: The words Calm/Windy, Dry/Rainy and Small/Bigg should be referred to in the figure caption.
18. Line 526: It is a bit confusing to use intensity here, since one of the clusters is also called intense.
19. Line 534 and 539: How are the cyclones in the XWS storm catalogue identified.  If they use SSI then it is not surprising that a large number of the SSI cluster are contained in the XWS storm catalogue. Similarly, the named storms are those that lead to impact, hence they are biased towards landfalling storms. Can these impact-based metrics be used independently to verify the usefulness of the storm clustering technique?
20. Lines 585-595:  This is interesting information, but the authors make no link to the results in these studies to their study, so I'm not sure why this information is included?
21. Line 620-634: As above, it would be helpful if the authors could highlight the novelty of their results and how they build on the work in the previous studies described in this section. How is the cluster analysis a 'new perspective to the classification of ETC life cycles'? Could the authors be clear about what they are adding to the scientific literature?
22. Line 643: Given these limitations, could different criteria be used to identify Mediterranean cyclones?

Typographical errors
1. Line 166: Missing space before the bracket.
2. Line 167: Extra space after 2003.
3. Line 249: 'drawback' should be 'drawbacks'.

---

## Referee Comment (RC3)

*Review of manuscript egusphere-2024-1749 submitted to Natural Hazards and Earth System Sciences (NHESS)*

**Classification of North Atlantic and European extratropical cyclones using multiple measures of intensity**

by Joona Samuel Cornér et al.

**General comments:**

This manuscript presents a systematic classification of Extratropical cyclones (ECs) over the Euro-Atlantic region based on two types of intensity metrics: dynamical and impact-based. The authors highlight using of 5 metrics to assess the intensity of ECs and present a classification of 4 clusters consistent with previous studies. I consider the present manuscript well-written, the analysis well-performed, and the topic of interest to the community of NHESS. Therefore, the manuscript may be published. Minor issues have been found that need to be addressed before the manuscript can be published. My detailed comments are found below.

**Specific comment:**

I found it novel that this study considers both the dynamics and impacts of cyclones. However, weak cyclones are located in the Mediterranean area where we have seen several EC damages in recent years. You have mentioned that Medicane Apollo is one of the weak cyclones (line 553). How would you suggest using your framework in operational forecasting to determine tropical cyclone impacts? (as suggested in lines 690-692) Is it necessary to look at all 5 metrics? If so, what role do the 4 clusters play?

**Technical corrections:**

-I found the abbreviation for 850 hPa relative vorticity ("VO") confusing. Why is not "RV850" used?

-For the case study, it might help to have a map with the cyclone tracks.

-The caption of Figure 2 needs more details.

-Line 16: What do you mean by "sensible weather"?

-Line 18: "Buildings" are also "infrastructure."

---

## Author Comment (AC1)

**Authors' Response to Reviewer 1**

> **General Comments.** This study uses ERA5 reanalyses for 43 extended winters to classify North Atlantic and European extratropical cyclones according to their intensity, based on multiple dynamical and impact-related measures. They perform a principle component analysis and a cluster analysis to find four clusters of extratropical cyclones that differ in terms of intensity, geographical location and characteristics. Cluster HighSSI is associated with the highest mean values in many intensity measures (e.g., low-level vorticity and wind speed), followed by cluster Intense. Despite a relatively small occurrence frequency (<10% of all cyclones), HighSSI contains a large fraction of well-known impactful storms. Most cyclones belong to cluster AvgMST (~44%). They have average intensities and occur preferentially over the main North Atlantic storm track. Cyclones in cluster Weak occur in about 25% of all cases, have relatively weak intensities and mainly occur in the Mediterranean and over Europe.
>
> Overall, the manuscript is interesting to read, the storyline is clear, the methods are appropriate, and the general conclusions are sound. Below are a few minor comments and suggestions.

**Response:** We would like to thank you for the feedback and the valuable comments which helped improve the quality of our manuscript. We have carefully addressed all the issues item by item as follows.

**Specific comments:**

**Comment 1**

The paper is rather long. This is not a criticism in itself, as the paper is well-written and interesting. However, maybe you can consider shortening the description of the datasets and methods a bit, such that the reader does not have to wait until page 11 for the results. For instance, although nicely written, it might not be necessary to mention all advantages and disadvantages of ERA5, which is a well-known and widely used reanalysis dataset. And in the result section you might consider moving Fig. 9 and its description to the supplement, to shorten the manuscript a bit (but I would also be fine if you left it in the main part).

**Response:**

We have shortened the data section including the description of ERA5 advantages and disadvantages. Individual sentences have been revised for the sake of brevity and clarity. We have also combined the discussion of Figure 9 with that of Figure 8 to reduce the amount of text. However, we feel that it is necessary to include Figure 9 in the main text as it shows details which are not evident in Figure 8 (e.g. almost equal proportion of cluster HighSSI ETCs in each area and large proportion of cluster AvgMST ETCs in Europe). These reductions in text partly compensate the revisions made based on reviewer comments, but overall the length of the paper has remained unchanged. See also related Specific Comment 2 of Reviewer 1 and Minor Comments 8, 9, and General Comments of Reviewer 4.

**Comment 2**

Abstract and conclusions: Most of the abstract describes your method, but it would be nice if you could add some sentences about your results (e.g., the percentage of cyclones in each of the four clusters and the differences between the clusters in intensity, geographical location and characteristics, etc.). Similarly in the conclusions. As the paper is quite long, a summary of the main results would be helpful.

**Response:**

A few sentences have been added to the abstract to include more details about the results. The part of the abstract now reads:

> The cluster analysis is able to produce four clusters between which cyclones differ in terms of their intensity, life cycle characteristics such as deepening rate and lifetime, and geographical location. The on average second most intense cluster contains around a fifth of all cyclones which occur mostly at the start of the North Atlantic storm track. Average intensity cyclones comprise nearly half of all cyclones and occur mostly at the northeastern parts of the storm track. Fourth of all cyclones belong to the cluster of on average the weakest cyclones which are found mostly over Europe and in the Mediterranean. The on average intense cyclones constitute less than a tenth of all tracks and occur almost equally everywhere. Based on average magnitude, most intensity measures are arranged in the same order between the clusters. There is also a link between average intensity and deepening rate, lifetime, and mean propagation speed of cyclones in the clusters.

The conclusions have been expanded by replacing the sentence

> One cluster has on average weak ETCs (cluster Weak), another one average ones (cluster AvgMST), and two clusters more intense ones (clusters Intense and HighSSI), with the most overlap in the distributions of the intensity measures in

the latter two.

with

Cluster Weak has on average the weakest ETCs (25.54 % of all ETCs), cluster AvgMST contains average intensity ETCs (44.42 %), and clusters Intense and HighSSI are composed of more intense ETCs (21.46 % and 8.57 %, respectively). Based on average magnitudes, most of the intensity measures have the clusters in the same order. However, the clusters are not discrete in the feature space defined by the intensity measures since there is overlap in the distributions of the intensity measures between the clusters. The most overlap is between clusters Intense and HighSSI.

and by adding the sentence

The average intensity of ETCs in the clusters can be qualitatively linked to their deepening rate, lifetime, and mean propagation speed.

Although we have shortened some sections of the text (see Comment 1 of Reviewer 1), these additions contribute to the text not being shorter than before revision.

**Comment 3**

Fig. 8a-d: Please add the units to the colorbar.

**Response:**

The colourbar label has been changed from "Difference with full climatology" to "Difference in number of tracks".

**Comment 4**

Fig. 10: Do I understand correctly that the positive trend in the number of intense cyclones and the negative trend in the number of weak cyclones indicates that the cyclones moved from the weak to the intense cluster because of the increase in cyclone-related precipitation?

**Response:**

If we understand the question correctly, yes, this is essentially the effect. Cluster Intense ETCs have on average larger precipitation values than cluster Weak ETCs. Precipitation values are on average higher towards the end of the time series, which we think is the main reason more Intense ETCs and less Weak ETCs occur at the end of the time series compared to the beginning.

**Comment 5**

Section 4: To define the clusters, you use four wind-related measures (WS850, VO, WFP and SSI), but only one measure for precipitation. Is this the reason why the cluster analysis discriminates intensity more based on wind than precipitation (see for instance Fig. 6b, c), and that cyclones with heavy precipitation but weak winds like Apollo belong to cluster weak rather than Intense? Is this a weakness of the method?

**Response:**

Thank you for your important comment. To answer your first question, we think that the stronger discrimination based on wind-related measures rather than precipitation is more linked to the shapes of the measures' distributions than the number of measures. Precipitation has smaller variance than WS850, VO or WFP (but higher than SSI) so it is more "difficult" for the cluster analysis to create non-overlapping clusters in terms of precipitation. For Apollo, we believe in a complementary explanation. The region of

interest encompassed the eastern coast of North America and the open ocean, two areas with larger storm-associated precipitation than in Europe (Hawcroft et al., 2012). As a result, the average precipitation (as depicted in section 4.2) is relatively high for Europe. This also explains why most of the XWS storms fall in the center of PC2 (see Figure 11). This explanation has been added to the discussion.

To answer your second question, the entry feature set (WS850, VO, WFP, SSI and PRECIP) has been chosen for interpretability purposes and not for efficiency. Because, the redundancy in wind-related features would be mainly considered as a drawback in a Machine Learning sense (Yu and Liu, 2004). This means that they don't individually create very much further discrimination between the clusters. Having said that, the method is inherently weak in this sense, as we state in line 565.
* * *
**Comment 6**

Section 4: I find it interesting that many of the well-known storms have below-average precipitation (Fig. 11). You say that this is because you picked many of the storms from the XWS catalogue that uses wind-based measures. I know the paper is already rather long, but maybe you can consider including a few more European cyclones that led to heavy precipitation events. It would be interesting to see whether they are located close to Apollo or rather somewhere else in the phase space.
* * *
**Response:**

We further note that the below-average precipitation in many of the European cyclones can be explained by the average precipitation value being affected by many cyclones occurring over the ocean, which makes the value larger compared to many storms occurring over land as discussed in Comment 5 of Reviewer 1. An explanation of this has been added to the discussion in the paper.

We agree with the reviewer's comment that it would be interesting to investigate more

well-known storms. However, there are quite many storms in the analysis and, as the reviewer noted, the paper is already rather long. Thus, we prefer leaving it for future work. We have added a short example script to the Zenodo repository with the trained sPCA included (Cornér et al., 2024). A reader can use this model and their own data (or our data) to see to where any storm which can be described with the 11 intensity measures falls in the sPCA space. A script has been provided which does the same for the GMM as well. This script predicts the clusters of objects which can be described with the reduced set of intensity measures, i.e., VO, WS850, PRECIP, WFP, and SSI. Mentions of these models have been added to the text as well. The use of the trained models is discussed also in General Comment 1 of Reviewer 2.
* * *
**Comment 7**

Section 4, line 558: "Moreover, ETCs which do have high precipitation values over land areas occur mostly over North America (not shown)." Where do you get this statement from, is it from your dataset? Over Europe ETCs can also produce substantial amounts of precipitation.

**Response:**

The statement is true for ETCs in our dataset. The sentence has been reformulated to convey this information more clearly. Another modification to the sentence has been made to indicate that high precipitation values are not restricted to only North America, but that in our dataset the highest values on average occur there:

> Moreover, in our dataset ETCs which do have high precipitation values over land areas occur mostly over North America (not shown).

This is in agreement with Hawcroft et al. (2012), who show that the east coast of North America receives more precipitation than Europe and proportionally more of it is associated with ETCs. This is also consistent with the fact that many European storms

do not seem to have high precipitation values, which is discussed in Specific Comment 5 of Reviewer 1. This is now clarified in the discussion in the paper.

**Typos and wording:**
* * *
**Comment 8**

Line 79: "Section 4 contains .. and Section 6 concludes . . . " Section 5 is not mentioned.
* * *
**Response:**

The sentence has been modified to include a mention of each section.
* * *
**Comment 9**

Line 118: "small scale" – maybe better: "small-scale"
* * *
**Response:**

We agree with the reviewer's comment and changed the word accordingly.
* * *
**Comment 10**

Lines 135: "beginning from" should be "beginning at"
* * *
**Response:**

We agree with the reviewer's comment and changed the expression accordingly.

**Comment 11**

Line 148: Should it be "obtain" instead of "contain"?

**Response:**

We agree with the reviewer's comment and changed the word accordingly.

**Comment 12**

Line 151: "... and are available as in the reanalysis." The wording is a bit cumbersome, maybe you can reformulate the sentence.

**Response:**

The sentence has been reformulated from

> and are available as is in the reanalysis.

to

> and can be obtained from the reanalysis with no or minimal post-processing.

**Comment 13**

Fig. 1: The abbreviation "ONDJFM" has not been introduced. It would fit in the last sentence of Section 2.1.

**Response:**

Since the abbreviation was used only once in the manuscript, we have removed it and replaced it with "October–March".

**Comment 14**

Line 167: remove empty space between double parentheses.

**Response:**

The empty space has been removed.

**Comment 15**

Lines 210, 214, 215: "in (each) grid point" should be "at (each) grid point"

**Response:**

The prepositions have been changed from "in" to "at" in the mentioned lines (also in line 188).

**Comment 16**

Line 211: "... whereas Leckebusch et al. (2008a) ... " This part of the sentence is difficult to understand, consider reformulating it. Is it actually necessary?

**Response:**

We agree with the reviewer that the explanation of the SSI used by Leckebusch et al. (2008) is not necessary and have thus removed this from the text.

**Comment 17**

Line 234: "For PRECIP, the chosen time is 12 hours before the time of maximum VO, for the same reason." Can you be a bit more specific and briefly motivate why you investigate PRECIP 12h before the time of maximum VO, i.e., at a different time than the other variables? (It does make sense to me, as the peak in precipitation often occurs before the peak in VO, and I guess you suggest this when you write "for the same reason", but the sentence is a bit vague.)

**Response:**

The sentence has been made clearer. What previously was

> For PRECIP, the chosen time is 12 hours before the time of maximum VO, for the same reason.

has been changed to

> The maximum precipitation rate occurs on average 12 h before time of maximum VO (see Fig. S5f). For this reason, PRECIP is evaluated at this time step.

**Comment 18**

Line 296: "all 11 of the intensity measures" – I would write "all 11 intensity measures"

**Response:**

The expression has been revised accordingly.

**Comment 19**

Table 1: The description of the column "Distance" could be more precise in the caption.

**Response:**

The description has been made more precise and now reads:

the maximum distance from the VO maximum (in geodesic degrees) to which the values are searched for

**Comment 20**

Caption Fig. 2: "times in the tracks" – I would write "times along the tracks"

**Response:**

The expression has been modified accordingly.

**Comment 21**

Caption Fig. 3: "An absolute value of Pearson's r is shown for correlations involving MSLPa to aid comparison with other coefficients." I don't know what you are referring to (I don't see anything that is different for MSLPa compared to the other measures).

**Response:**

The correlation between MSLPa and e.g. VO is negative as MSLPa decreases for a stronger storm whereas VO increases. The sentence lacked information and has now been made more explicit from

> An absolute value of Pearson's r is shown for correlations involving MSLPa to aid comparison with other coefficients.

to

> All values of Pearson's $r$ involving MSLPa are negative, but an absolute value is shown for them to aid comparison with other coefficients.

**Comment 22**

Line 474: "Atlantic" – I would write "North Atlantic"

**Response:**

The expression has been changed accordingly. It has also been changed in the caption of Figure 9 and in the text discussing the figure to ensure consistency.

**Comment 23**

Line 489: Do you mean the southeastern instead of the northeastern coast?

**Response:**

The word has been changed to "eastern" to be more general.

**Comment 24**

Line 490: "... storm track, and elsewhere ..." I would split the sentence into two: "... storm track. Elsewhere ..."

**Response:**

The sentence has been modified accordingly.
* * *
**Comment 25**

Line 496: consider reformulating the sentence to "As in cluster Intense, the highest track densities occur at the start of the storm track." The second part of sentence I would leave away, as the values are increased everywhere in the main storm track and not only at the beginning.

**Response:**

The sentence has been modified accordingly.
* * *
**Comment 26**

Line 499: What do you mean by "discontinuous"? Consider reformulating the sentence.

**Response:**

Discontinuous in this context means that the distribution of track density has multiple local maxima as opposed to a single one. The clusters other than HighSSI have their extreme track density values in certain locations from which the values increase/decrease continuously. To briefly clarify this in the text, the sentence has been reformulated from

> the values in this area are more discontinuous, which is likely due to the effect of normalization.

to

the values in this area are more discontinuous with multiple local maxima, which is likely due to the effect of normalization.

**Comment 27**

Caption Fig. 6: "The percentages in parentheses indicate how large proportion of all tracks is in each cluster." Better: "... indicate the proportion of all tracks in each cluster"

**Response:**

The sentence has been modified accordingly.

**Comment 28**

Line 502: "over Europe"

**Response:**

The expression has been changed accordingly.

**Comment 29**

Line 586: "likened" – Typo: "linked"

**Response:**

The expression has been changed accordingly.

**Comment 30**

Line 591: "performed analysis" should be "performed an analysis" or "performed analyses"

**Response:**

The expression has been changed to "performed an analysis".

**Comment 31**

Line 608: Maybe you can briefly mention what the Petterssen and Smebye types A and B are.

**Response:**

A sentence explaining the features of these ETC types has been added:

Type A ETCs are associated with development in a baroclinic zone with weak upper-level and strong lower-level forcing. Type B ETCs develop on a pre-existing upper-level trough with strong upper-level forcing and initially weak lower-level forcing.

**Comment 32**

Line 665: "correlation of coefficient" should be "correlation coefficient"

**Response:**

The expression has been changed accordingly.

**Comment 33**

Lines 662 and 669: I would change "had" to "has" and "were" to "are" to consistently use present tense.

**Response:**

The tense has been made consistent accordingly.

**Comment 34**

Line 666: "This, along with the fact that a much larger proportion of named impactful storms than all ETCs belong to cluster HighSSI, ..." Complicated wording, consider reformulating it.

**Response:**

The sentence has been reformulated from

> This, along with the fact that a much larger proportion of named impactful storms than all ETCs belong to cluster HighSSI

to

> This, along with the fact that a large majority (81 %) of named impactful storms belong to cluster HighSSI despite the cluster accounting for less than 10 % of all ETCs

> ### Comment 35
>
> Line 671: "... but HighSSI ETCs are almost as frequent everywhere, given the distribution of ETC occurrence in the domain." This part of the sentence is a bit difficult to understand, consider reformulating it.

**Response:**

The expression has been changed from

> given the distribution of ETC occurrence in the domain

to

> when normalized with respect to the climatological distribution of ETC occurrence

**References**

M. K. Hawcroft, L. C. Shaffrey, K. I. Hodges, and H. F. Dacre. How much Northern Hemisphere precipitation is associated with extratropical cyclones? 39(24), 2012. doi: 10.1029/2012GL053866.

Lei Yu and Huan Liu. Efficient feature selection via analysis of relevance and redundancy. *The Journal of Machine Learning Research*, 5:1205–1224, 2004.

Joona Cornér, Clément Bouvier, Benjamin Doiteau, Florian Pantillon, and Victoria A. Sinclair. *Classification of extratropical cyclones using multiple measures of intensity: Data and Python code*, September 2024. URL `https://doi.org/10.5281/zenodo.11384417`.

Gregor C. Leckebusch, Dominik Renggli, and Uwe Ulbrich. Development and application of an objective storm severity measure for the Northeast Atlantic region. 17:575–587,

2008. doi: 10.1127/0941-2948/2008/0323. URL https://api.semanticscholar.org/CorpusID:123018345.

---

## Author Comment (AC2)

**Authors' Response to Reviewer 2**

**General Comments.** This paper aims to produce cyclones classes which capture various aspects of cyclone intensity better than metrics that use a single diagnostic. Overall, the paper describes the motivation and methods used well. There are however several parts of the paper that require clarification, particularly the clustering method, and more work is required to illustrate how future studies can implement the recommendations made. These points are described below in more detail and should be addressed before the paper is suitable for publication in NHESS.

**Response:** We would like to thank you for the feedback and the valuable comments which helped improve the quality of our manuscript. We have carefully addressed all the issues item by item as follows.

**Comment 1**

My main concern is that the 4 clusters identified by the authors are not straightforward to implement by others in future studies. The authors recommend that 5 variables are considered, but how to combine these to identify different classes of cyclone in future studies is not clear. Will subsequent studies need to repeat the Gaussian mixture model with their own tracked cyclone data to identify the 4 clusters? Also, the clusters are identified from the sPCA figure, so that step would also need to be repeated I believe. I would like the authors to provide a more step-by-step guide to how the intensity measures should be combined to 'comprehensively and non-redundantly quantify the intensity of ETCs' (lines 645 and 659). If the cyclone clusters are to be used to see how different kinds of ETCs respond to climate change (line 688) simpler instructions are needed on how to create them.

On a related point, on line 692 the authors state that their method 'allows a vast amount of information to be condensed to a level that is manageable for operational forecasters', how would the forecasters use the information? Is it envisaged that they would be provided with the 'intensity' of a cyclone based on a score from each of the clusters, or do the authors have something else in mind?

**Response:**

To address the issues about reproducibility and applicability of our analysis to other datasets, we have added a short example script to the Zenodo repository with the trained sPCA model included. A reader can use this model and their own data (or our data) to see to where any storm or object which can be described with the 11 intensity measures falls in the sPCA space (Cornér et al., 2024). A script has been provided which does the same for the Gaussian mixture model (GMM) as well. This script predicts the clusters of objects which can be described with the reduced set of intensity measures, i.e., VO, WS850, PRECIP, WFP, and SSI. Mentions of these models have been added to the text as well. We propose that this set of five intensity measures is "comprehensive and

non-redundant", i.e. it describes the intensity from all relevant aspects and does not contain the same information twice. These measures cannot be combined in the literal sense of the word, but should be used together when quantifying ETC intensity. For example, we claim that investigating only vorticity, one cannot draw conclusions about the increase or decrease of ETC intensity.

The clusters are not identified from the sPCA figure. The clusters are produced with the GMM which is run with the reduced set of intensity measures. This reduced set is identified by using the sPCA result and correlations between the intensity measures. This has been clarified in the text.

To answer the second part of the comment: Our framework could be used in operational forecasting to assess the uncertainty of possible storm impacts. Although we reduce the set of intensity measures to five measures, this is a lot of information to investigate when one uses an ensemble prediction system with possibly tens of ensemble members. We suggest that the information could be condensed to a manageable level by first determining the cluster of a storm in each ensemble member by using the trained GMM instance and then seeing how much disagreement there is in the ensemble. This would offer information on both the intensity and possible impacts of a storm as well as how uncertain this estimate is. This of course requires additional information such as predictions of storm locations and is dependent on the selection of points of interest.

**Specific comments:**

> ### Comment 1
> Line 11: What do the authors mean by 'impactful storms'?

**Response:**

By impactful storms we refer to extratropical cyclones which had a societal impact due

to e.g. heavy precipitation-associated flooding or other damage to infrastructure from winds.

**Comment 2**

Line 35: Vorticity is a noisy field typically including both mesoscale and synoptic scale features. Are the authors referring to a filtered vorticity field when they say that vorticity metrics describe the synoptic scale dynamics of ETCs?

**Response:**

This does in fact refer to filtered vorticity which represents synoptic-scale features. The filtering is explained later in the text in Section 2.2. We have decided to keep the text in the introduction unchanged as we do not want to introduce technical details in it.

**Comment 3**

Line 46 and 54: What is meant by 'concise' metrics? If the aim is to produce a concise metric, this should be defined.

**Response:**

We have revised the sentence in the text to clarify what is meant by concise metrics as follows:

> Secondly, a manageable number of metrics which are easy to compute (concise metrics) are needed to identify whether any trends in ETCs intensity have already occurred or may do in the future as the climate changes.

We also want to highlight that producing a single concise metric is not an aim of the paper and therefore avoid using the singular form of the word in this context.

> ## Comment 4
>
> Line88: What is T1639?

**Response:**

$T_L639$ refers to the spectral truncation of the grid in model which has been used to produce ERA5 and thus the output resolution of ERA5. It is a linear triangular truncation in which the largest total wavenumber that can be represented in the grid is 639. See explanation e.g. here: https://confluence.ecmwf.int/display/OIFS/4.3+OpenIFS%3A+Horizontal+Resolution+and+Configurations.

> ## Comment 5
>
> Line 102: What is the consequence of ignoring the biases in ERA5 data? Are you results sensitive to these biases?

**Response:**

When using a reanalysis, it is impossible to avoid biases in the data. All reanalysis datasets have some biases and choosing between reanalyses means choosing between biases. We determined that ERA5 is the best suited for our study in the North Atlantic–European region as it has been shown to perform well in many aspects related to ETCs and is easily available.

It is difficult to estimate the consequence of ignoring biases or the sensitivity of the results to these. We can speculate that the underestimation of high precipitation values in ERA5 may cause our precipitation distribution to be too narrow. This may have an effect on the cluster analysis through e.g. creating more overlap in precipitation between the clusters. We have added a mention of this is the discussion. It would perhaps be beneficial to perform similar analysis with a different reanalysis dataset but it is outside the scope of this paper.

**Comment 6**

Line 125: What are the time steps referred to here?

**Response:**

The time steps refer to the time steps in the data. Two days equals 16 time steps with 3-hourly data. This has been clarified in the text.

**Comment 7**

Line 151: '... available as is in the reanalysis.' This sentence does not make sense to me.

**Response:**

The comment has been answered also in Comment 12 of Reviewer 1 which states: The sentence has been reformulated from

> and are available as is in the reanalysis.

to

> and can be obtained from the reanalysis with no or minimal post-processing.

**Comment 8**

Line 163 and 177: What is the consequence of a mismatch or even no match between the location of the vorticity maxima and mslp minima? Is the latter a consequence of the fact that vorticity can capture the early stages of cyclone development before a closed isobar is identified in mslp?

**Response:**

The consequence of a mismatch between the location of vorticity and MSLP extrema is that the wind field is investigated around the vorticity maximum whereas traditionally the closed MSLP minimum might be considered as the reference for an ETC centre. However, in most cases the distance between these two extrema is smaller than the distance between the vorticity maximum and the wind speed maxima, which means the associated wind speed values are largely unaffected. We do not believe that the latter is a consequence of the lack of a closed isobar. The MSLP minimum does not need to be closed but just needs to be a local minimum.

A sentence has been added to clarify how the tracks with no found associated MSLP value are dealt with:

> For a small number (3 %) of maximum VO values, TRACK is unable to find an associated MSLPa value. These ETCs are omitted from the dataset.
* * *
**Comment 9**

Line177, 330: Here and elsewhere the authors refer to ERA5 wind gusts. It would be useful to have a brief explanation of this diagnostic quantity and how it is derived? Why is it underestimated in some areas (line 206)?
* * *
**Response:**

Wind gusts in ERA5 are calculated as the sum of the 10 m wind speed, a term accounting for surface roughness, and a term representing the contribution of convective downdrafts (Equation 3.99 in ECMWF, 2016). Wind gusts are underestimated in some areas because one or more of these terms is underestimated. The underestimation occurs mostly in regions of complex orography and/or areas with high wind speeds (Chen et al., 2024; Minola et al., 2020). This is briefly discussed in the Discussion section of the paper.

The details of ERA5 wind gust and how we use it were originally included in the manuscript but were removed for the sake of shortening the text. A reference of how wind gust is calculated in ERA5 is included in the paper (Bechtold and Bidlot, 2009).

**Comment 10**

Table 1: What is the difference between accumulated and time-integrated?

**Response:**

This is briefly explained at the end of section 2.3.2. The precipitation values are pre-processed by summing together 1-hourly values to obtain an accumulated precipitation rate per 3 hours. Therefore, we can just sum the values together to get an accumulation throughout the whole ETC track. The SSI values are, however, represented only every 3 hours as instantaneous values and cannot be summed together in the same way as precipitation values that are accumulations. Instead of calculating the accumulation by summing together the instantaneous values every three hours, we integrate them with respect to time, i.e. multiply the difference between respective time steps with a time interval of 3 hours. This ensures that the accumulated SSI value is (nearly) independent of the temporal frequency the data are available at. We use the word "accumulated" for both SSI and precipitation since the interpretation of both measures is similar: they quantify the relevancy for impact across the whole track.

**Comment 11**

Line 252: Do the authors use the 'type' of correlation, i.e. linear or non-linear later in the analysis or interpretation of their results? I may have missed this. Does the fact that the MI correlations are higher for SSI than the Pearson correlation imply that they are non-linearly related to the windspeed for example?

**Response:**

The fact that the correlation between the dynamical intensity measures and SSI is non-linear, is used to justify the inclusion of SSI in the reduced set of comprehensive intensity measures. The reasoning behind this is that as opposed to the wind-related measures, which are strongly correlated with each other, the non-linear relationship between them and SSI means that SSI is able to possibly create more separation in the feature space between the clusters. The fact that MI correlations are higher for SSI than the Pearson correlation does in fact imply that the relationship to e.g. wind speed is non-linear. This is mentioned at the end of section 4.1.
* * *
**Comment 12**

Line 255: Why is it important to know that the method is 'heavily' used? This does not imply that it is the most appropriate method for this study.
* * *
**Response:**

The word has been deleted from the text as it was deemed unnecessary.
* * *
**Comment 13**

Lines 281: What is the silhouette score?
* * *
**Response:**

The silhouette score measures the proximity of samples in other clusters to a sample in a specific cluster in the feature space. The silhouette score is the better (closer to 1) the closer a sample is to its own cluster's centroid than the nearest cluster's centroid that the sample is not a part of. Essentially, it tells how distinct the clusters are from one another. See Shahapure and Nicholas (2020) for further explanation. This reference has been added to the text as well.

**Comment 14**

Figure 2 caption: The caption should refer to table 1 for details of the intensity measures.

**Response:**

The figure caption has been modified to include this reference.

**Comment 15**

Line 365: Why do the SSI measures have no weight? What is the interpretation of this result?

**Response:**

The SSI measures have no weight in the Sparse PCA because they have very small variances compared to the other measures. Many ETCs have very small or zero SSI and only some have moderate or large SSI (the distribution is far from Gaussian even on a base-10 logarithmic scale). This causes the SSI variances to be much smaller than those of other intensity measures. The interpretation is that the SSI measures do not represent the variability of the whole dataset very well.

**Comment 16**

Line 395: Here the 4 clusters and their names are introduced. Are the clusters identified from the sPCA figure (fig 5)? Or have I misunderstood the methodology here? Also, since these names are used frequently in the remainder of the paper, I would suggest using bullet points so that they stand out in the text.

**Response:**

The clusters are not identified from the sPCA figure. Instead, they are identified by using the Gaussian mixture modelling (GMM) method which is introduced in Section

3.3. The GMM is performed with the identified intensity measures, which come mainly from the sPCA, as input. The use of the methodology has been clarified in the text by reformulationg the sentence to

> The cluster analysis was performed using the method described in Sect. 3.3 with the reduced set of intensity measures identified in Sect. 4.2 as input.

See also General Comment 1 of Reviewer 2.

We have changed the listing of the clusters to include bullet points. Thank you for the suggestion.

**Comment 17**

Figure 5: The words Calm/Windy, Dry/Rainy and Small/Big should be referred to in the figure caption.

**Response:**

The figure caption has been modified to include the sentence

> The labels Calm & Windy, Dry & Rainy, and Small & Big refer to the qualitative interpretation of PC1, PC2, and PC3, respectively.

**Comment 18**

Line 526: It is a bit confusing to use intensity here, since one of the clusters is also called intense.

**Response:**

We have not modified this sentence as throughout the manuscript we have discussed and referred to the intensity of extratropical cyclones and therefore do not think this is confusing. Additionally, we want to keep our terminology consistent.

**Comment 19**

Line 534 and 539: How are the cyclones in the XWS storm catalogue identified. If they use SSI then it is not surprising that a large number of the SSI cluster are contained in the XWS storm catalogue. Similarly, the named storms are those that lead to impact, hence they are biased towards landfalling storms. Can these impact-based metrics be used independently to verify the usefulness of the storm clustering technique?

**Response:**

The named storms in the XWS catalogue were chosen based on the amount of insured loss they caused. While one aim for the creators of the catalogue was to find an index which would rank the named storms highly and this index is similar to the SSI we used, the storms were not selected by using this index. Our result is consistent with theirs in this regard: SSI ranks these named storms which caused large insured losses highly.

We have not tested whether the impact-relevant metrics can be used independently to verify the usefulness of the cluster analysis. Of the impact-relevant metrics, SSI seems to have a large effect on the cluster to which storms get assigned. We have however shown this for only a small sample of high-impact storms so the result cannot be generalized. Furthermore, all investigated storms in the HighSSI cluster do not have large values in all of the impact-relevant metrics. For example, storm Christian/St. Jude belongs to cluster HighSSI but has below average values in PC2 and PC3, i.e. precipitation and wind footprint. However, storm Christian has above average wind speeds (not impact-relevant by our definition) which affect the determination of its cluster as well.

**Comment 20**

Lines 585-595: This is interesting information, but the authors make no link to the results in these studies to their study, so I'm not sure why this information is included?

**Response:**

We have extended the discussion of this result from

> Their northward-oriented tracks were mostly found near the eastern coast of North America, which means it could contain many of the same tracks as in our clusters HighSSI or Intense (e.g., post-tropical cyclones). In terms of MSLP, their northward-oriented ETCs were the most intense but the northeastward-oriented ETCs were faster-moving, while they found no significant differences in ETC lifetime between any of the clusters.

to

> Their northward-oriented cluster tracks were mostly found near the eastern coast of North America and were among the most intense ETCs in terms of MSLP. This indicates that the northward-oriented tracks could contain many of the same tracks as our clusters HighSSI or Intense (e.g., post-tropical cyclones). They also found that the northeastward-oriented ETCs were the fastest-moving, while they found no significant differences in ETC lifetime between any of the clusters. This result is different from ours, as we found a link between average dynamical intensity and ETC speed and lifetime.

**Comment 21**

Line 620-634: As above, it would be helpful if the authors could highlight the novelty of their results and how they build on the work in the previous studies described in this section. How is the cluster analysis a 'new perspective to the classification of ETC life cycles'? Could the authors be clear about what they are adding to the scientific literature?

**Response:**

The paragraph has been revised to be more clear about the stated claims and now reads:

While these types of analyses are suitable for studying the precursors and forcing mechanisms of ETCs, we demonstrate that classification of ETCs based on their intensity benefits from an added level of objectivity via the cluster analysis. This can be seen in the overlap between the intensity measure distributions for different clusters in Fig. 6. Despite this overlap introduced by the objective method, our clusters can be at least qualitatively linked to classes of ETCs obtained with more subjective methods as described above. In fact, a possible course of future study is the identification of the variability in the ETC precursors and forcing mechanisms within our clusters. We believe that this form of analysis, which links the intensity and relevance for impacts to the genesis environment of ETCs, would offer a new perspective to the classification of ETC life cycles and possibly improve the predictability of ETC intensity.

**Comment 22**

Line 643: Given these limitations, could different criteria be used to identify Mediterranean cyclones?

**Response:**

We agree that different criteria could be used to identify Mediterranean cyclones better. However, this would likely introduce inconsistency and more subjectivity compared to the Atlantic cyclones identified with the criteria. Although we acknowledge the limitations in the criteria used, we show that the number of tracked Mediterranean cyclones is not very different from literature.

**Typographical errors:**

> **Comment 1**
>
> Line 166: Missing space before the bracket.

**Response:**

A space has been added before the bracket.

> **Comment 2**
>
> Line 167: Extra space after 2003.

**Response:**

The empty space has been removed.

> **Comment 3**
>
> Line 249: 'drawback' should be 'drawbacks'.

**Response:**

The word has been changed accordingly.

**References**

Joona Cornér, Clément Bouvier, Benjamin Doiteau, Florian Pantillon, and Victoria A. Sinclair. *Classification of extratropical cyclones using multiple measures of intensity: Data and Python code*, September 2024. URL `https://doi.org/10.5281/zenodo.11384417`.

ECMWF. *IFS Documentation CY41R2 - Part IV: Physical Processes*. Number 4. ECMWF, 2016 2016. doi: 10.21957/tr5rv27xu. URL `https://www.ecmwf.int/node/16648`.

Ting-Chen Chen, François Collet, and Alejandro Di Luca. Evaluation of ERA5 precipitation and 10-m wind speed associated with extratropical cyclones using station data over North America. 2024. doi: 10.1002/joc.8339.

Lorenzo Minola, Fuqing Zhang, Cesar Azorin-Molina, AA Safaei Pirooz, RGJ Flay, Hans Hersbach, and Deliang Chen. Near-surface mean and gust wind speeds in ERA5 across Sweden: towards an improved gust parametrization. 55(3-4):887–907, 2020. doi: 10.1007/s00382-020-05302-6.

Peter Bechtold and Jean-Raymond Bidlot. Parametrization of convective gusts, 2009 2009. URL `https://www.ecmwf.int/node/17487`.

Ketan Rajshekhar Shahapure and Charles Nicholas. Cluster quality analysis using silhouette score. In *2020 IEEE 7th International Conference on Data Science and Advanced Analytics (DSAA)*, pages 747–748, 2020. doi: 10.1109/DSAA49011.2020.00096.

---

## Author Comment (AC3)

**Authors' Response to Reviewer 3**

> **General Comments.** This manuscript presents a systematic classification of Extratropical cyclones (ECs) over the Euro-Atlantic region based on two types of intensity metrics: dynamical and impact-based. The authors highlight using of 5 metrics to assess the intensity of ECs and present a classification of 4 clusters consistent with previous studies. I consider the present manuscript well-written, the analysis well-performed, and the topic of interest to the community of NHESS. Therefore, the manuscript may be published. Minor issues have been found that need to be addressed before the manuscript can be published. My detailed comments are found below.

**Response:** We would like to thank you for the feedback and the valuable comments which helped improve the quality of our manuscript. We have carefully addressed all the issues item by item as follows.

**Specific comments:**

> ### Comment 1
>
> I found it novel that this study considers both the dynamics and impacts of cyclones. However, weak cyclones are located in the Mediterranean area where we have seen several EC damages in recent years. You have mentioned that Medicane Apollo is one of the weak cyclones (line 553).
>
> How would you suggest using your framework in operational forecasting to determine tropical cyclone impacts? (as suggested in lines 690-692) Is it necessary to look at all 5 metrics? If so, what role do the 4 clusters play?

**Response:**

As to why medicane Apollo is classified as "Weak", we believe in the following explanation. The region of interest encompassed the eastern coast of North America and the open ocean, two areas with larger storm-associated precipitation than in Europe (Hawcroft et al., 2012). As a result, the average precipitation (as depicted in section 4.2) is relatively high for Europe. This also explains why most of the XWS storms fall in the center of PC2 (see Figure 11). This explanation has been added to the discussion. As the cluster analysis is also dependent on the wind-based intensity measures, the above-average precipitation is not enough to classify Apollo to one of the on average more intense clusters. This has also been discussed in Specific Comments 5, 6, and 7 of Reviewer 1. These issues have also been addressed in the discussion of the revised manuscript.

To answer the second part of the comment: We assume the reviewer is asking about extratropical cyclone impacts rather than tropical cyclone impacts given the topic of our study. As discussed also in the second part of General Comment 1 of Reviewer 2, we state that our framework could be used in operational forecasting to assess the uncertainty of possible storm impacts. Although we reduce the set of intensity measures to five measures, this is a lot of information to investigate when one uses an ensemble prediction system with possibly tens of ensemble members. We suggest that the information could be condensed to a manageable level by first determining the cluster of a storm in each ensemble member by using the trained GMM instance and then seeing how much disagreement there is in the ensemble. This would offer information on both the intensity and possible impacts of a storm as well as how uncertain this estimate is. This of course requires additional information such as predictions of storm locations and is dependent on the selection of points of interest. We propose that in this framework it is necessary to look at all five metrics. However, by using the trained GMM instance, these five metrics can be condensed into a single number, a cluster label. With enough statistical power, i.e. in this case ensemble members, it is possible to assess the reliability of this label.

**Technical corrections:**

**Comment 1**

I found the abbreviation for 850 hPa relative vorticity ("VO") confusing. Why is not "RV850" used?

**Response:**

The abbreviation "VO" is used for relative vorticity in the data documentation of ERA5. Since vorticity is investigated only at a single level, the need for the specifying "850" was not considered necessary.

**Comment 2**

For the case study, it might help to have a map with the cyclone tracks.

**Response:**

A figure (Figure S7) which shows the tracks of the case study storms has been added to the supplement .

**Comment 3**

The caption of Figure 2 needs more details.

**Response:**

References to the section explaining the intensity measures and Table 1, which summarizes the measures, have been added to the figure caption.

**Comment 4**

Line 16: What do you mean by "sensible weather"?

**Response:**

By sensible weather we mean the weather that humans can sense, i.e., winds and rain.

To avoid unnecessary confusion, the word "sensible" has been removed from the sentence.

**Comment 5**

Line 18: "Buildings" are also "infrastructure."

**Response:**

The word "buildings" has been removed from the sentence.

---

## Author Comment (AC4)

**Authors' Response to Reviewer 4**

**General Comments.** The authors here present an interesting study where they examine variability and relationships between numerous measures of intensity of extratropical cyclones in the North Atlantic in ERA5. This is a very comprehensive study and methodical in nature and presents very interesting results. The main results of interest are on the clusters of storms, their different geographic locations and also different populations in the wider intensity distributions. Due to the methodical nature of this study, it clearly presents very worthwhile findings, my only criticism of this work is that the study is incredibly long and in some places perhaps overly so. I understand why this is as the authors have conducted a lot of analysis, however in places brevity may be an option. I have other minor comments, which i have detailed below, and once these have been addressed i think this study will be an excellent addition to this journal.

**Response:** We would like to thank you for the feedback and valuable comments which helped improve the quality of our manuscript. To address the concerns about the length of the study, we have shortened the data section and throughout the manuscript individual sentences have been revised for the sake of brevity and clarity. We have also combined the discussion of Figure 9 with that of Figure 8 to reduce the amount of text. These reductions in text partly compensate the revisions made based on reviewer comments, but overall the length of the paper has remained unchanged. We have carefully addressed all other issues item by item as follows.

**Minor comments:**

**Comment 1**

L37/38 "However, in many cases, neither the minimum MSLP nor maximum VO correlates well with the impacts of a given ETC" needs a reference

**Response:**

We have added the following references to the sentence: Field and Wood (2007); Roberts et al. (2014); Sinclair and Catto (2023).

**Comment 2**

L52 - specify that you are referring to "average strength" cyclones

**Response:**

The word "average" has been changed to "average-strength".

**Comment 3**

L88 - typo in the statement of spectral resolution

**Response:**

The statement has been changed to $T_L639$ to avoid confusion.

**Comment 4**

Section 2.2 - do you also perform the tracking using 3-hourly resolution, or use the standard 6-hourly approach? Please specify in this section

**Response:**

We have added the sentence

> The tracking is performed with 3-hourly VO data.

to make it clear that the ETC tracking is done with 3-hourly resolution.

**Comment 5**

L111 - I assume you are using VO at 850 hPa for tracking, but you do not state this initially, this needs to be done

**Response:**

We have added the sentence

> The tracking is performed with 3-hourly VO data.

to make it clear that the ETC tracking is done with 850 hPa relative vorticity.

**Comment 6**

L161 - your baseline intensity measure is VO from the tracks. If this is the case you need to state that this is the T42 filtered vorticity, and not vorticity that you would get directly out of ERA5. Anyone trying to re-produce your results would have to do the tracking in order to get this variable

**Response:**

The sentence has been changed from

> We use VO directly from the output of TRACK.

to

> We use VO at the T42 resolution directly from the output of TRACK.

to make it clear that vorticity is used at the T42 resolution.

The T42 filtered vorticity values in the TRACK output are available in the Zenodo repository (Cornér et al., 2024).

**Comment 7**

L245 - i believe there is a brackets issue in this line

**Response:**

The notation $[0,\infty)$ means that the 0 is a closed end, i.e. the value can be zero, while the $\infty$ is an open end, i.e. the value cannot be exactly $\infty$.

**Comment 8**

Section 4.3 - all sections here are very long and descriptive and i feel is an area where some text can be removed and sections shortened for the benefit of future readers.

**Response:**

Some text has been removed and paragraphs combined to shorten the section and the flow of the text has been improved. Individual sentences have been revised for the sake of brevity and clarity.

**Comment 9**

L500-511 - you state for this paragraph that the results of Fig 9 are the same as Fig 8, which is what you would expect as it is just the same data but presented in a different manner. If this paragraph (and Fig. 9) are not essential for your results i would consider removing.

**Response:**

We have combined the discussion of Figure 9 with that of Figure 8 to reduce the amount of text. However, we feel that it is necessary to include Figure 9 in the main text as it shows details which are not evident in Figure 8 (e.g. almost equal proportion of cluster HighSSI ETCs in each area and large proportion of cluster AvgMST ETCs in Europe). These reductions in text partly compensate the revisions made based on reviewer comments, but overall the length of the paper has remained unchanged. See also related Comment 2 of Reviewer 1.

**Comment 10**

Figure 10 - it would be good to state the trend value (and associated uncertainty) in each panel and the associated places in the text. Also, how much are your trends associated with modes of variability such as the NAO?

**Response:**

Equations for the trend values have been added to the figure and to the text, i.e. how much the number of ETCs per season change on average throughout the period. This trend value does not have an uncertainty associated with it but the p-value indicates how statistically significant the trend is.

According to Eade et al. (2022) the NAO index has been mostly positive within our study period from 1979 onwards and apart from a dip in the mid 1990's, the trend has been positive as well. This is qualitatively in line with our results which show less tracks

in Southern Europe and the Mediterranean (cluster Weak), a situation associated with a positive NAO index. It is also consistent with the results of (Laurila et al., 2021, Fig. 14) who found a positive correlation between the NAO index and 10 m wind speed in the North Atlantic and Finland, and a negative correlation in the Iberian peninsula. However, discrepancies arise with clusters Intense and AvgMST which both exhibit more tracks towards Northern Europe, a situation also associated with a positive NAO index. Their trends in number of ETCs as well as the signs of correlation with the NAO index are opposing for these two clusters (not shown). This discrepancy could be more related to the increase in precipitation in all ETCs rather than the location or orientation of ETC tracks. We of course need to keep in mind that the spatial distribution of ETCs in each cluster does not reveal their temporal variation and vice versa.

**References**

Paul R Field and Robert Wood. Precipitation and cloud structure in midlatitude cyclones. 20(2):233–254, 2007. doi: 10.1175/JCLI3998.1.

JF Roberts, AJ Champion, LC Dawkins, KI Hodges, LC Shaffrey, DB Stephenson, MA Stringer, HE Thornton, and BD Youngman. The XWS open access catalogue of extreme European windstorms from 1979 to 2012. 14(9):2487–2501, 2014. doi: 10.5194/nhess-14-2487-2014.

V. A. Sinclair and J. L. Catto. The relationship between extra-tropical cyclone intensity and precipitation in idealised current and future climates. 4(3):567–589, 2023. doi: 10.5194/wcd-4-567-2023. URL https://wcd.copernicus.org/articles/4/567/2023/.

Joona Cornér, Clément Bouvier, Benjamin Doiteau, Florian Pantillon, and Victoria A. Sinclair. *Classification of extratropical cyclones using multiple measures of intensity: Data and Python code*, September 2024. URL https://doi.org/10.5281/zenodo.11384417.

R. Eade, D. B. Stephenson, A. A. Scaife, and D. M. Smith. Quantifying the rarity of extreme multi-decadal trends: how unusual was the late twentieth century trend in the north atlantic oscillation? *Climate Dynamics*, 58(5):1555–1568, Mar 2022. ISSN 1432-0894. doi: 10.1007/s00382-021-05978-4. URL `https://doi.org/10.1007/s00382-021-05978-4`.

Terhi K. Laurila, Victoria A. Sinclair, and Hilppa Gregow. Climatology, variability, and trends in near-surface wind speeds over the North Atlantic and Europe during 1979–2018 based on ERA5. 41(4):2253–2278, 2021. doi: 10.1002/joc.6957. URL `https://rmets.onlinelibrary.wiley.com/doi/abs/10.1002/joc.6957`.